# Don't Just Fine-tune the Agent, Tune the Environment

**Siyuan Lu**[4,2,3,1*]  **Zechuan Wang**[1,4*]  **Hongxuan Zhang**[4,5]  **Qintong Wu**[4]
**Leilei Gan**[1†]  **Chenyi Zhuang**[4†]  **Jinjie Gu**[4]  **Tao Lin**[3,4†]

[1]Zhejiang University  [2]Shanghai Innovation Institute  [3]Westlake University
[4]Inclusion AI, Ant Group  [5]Nanjing University

## Abstract

Large Language Model (LLM) agents show great promise for complex, multi-turn tool-use tasks, but their development is often hampered by the extreme scarcity of high-quality training data. Supervised fine-tuning (SFT) on synthetic data leads to overfitting, whereas standard reinforcement learning (RL) struggles with a critical cold-start problem and training instability. To address these challenges, we introduce ENVIRONMENT TUNING, a novel training paradigm that enables agents to learn complex behaviors directly from problem instances without relying on pre-collected expert trajectories. ENVIRONMENT TUNING orchestrates this learning process through a structured curriculum, actionable environment augmentation that provides corrective feedback, and fine-grained progress rewards to ensure stable and efficient exploration. Using only 400 problem instances from Berkeley Function-Calling Leaderboard (BFCL) benchmark, our method not only achieves competitive in-distribution performance against strong baselines but also demonstrates superior out-of-distribution generalization, overcoming the performance collapse common to SFT-based approaches. Our work presents a paradigm shift from supervised fine-tuning on static trajectories to dynamic, environment-based exploration, paving the way for training more robust and data-efficient agents.

## 1 Introduction

The emergence of Large Language Model (LLM)-based agents, equipped with capabilities for intricate reasoning, planning, and tool interaction (Wang et al., 2024; Weng, 2023), has unlocked the potential to address complex, real-world problems across diverse domains like software engineering (Jimenez et al., 2023; Yang et al., 2024), computer use (Xie et al., 2024; OpenAI, 2025c; Wang et al., 2025a) and web browsing (Wei et al., 2025; OpenAI, 2025a; Moonshot AI, 2025). Much of this success has been demonstrated in single-turn tasks, such as mathematical reasoning, where plentiful datasets (Hendrycks et al., 2021; Sun et al., 2025), automated verifiers (Kydlíček, 2025), and advanced algorithms (Shao et al., 2024; Yu et al., 2025) have enabled significant progress (Zeng et al., 2025; He et al., 2025). Despite their utility, these agents, limited by a structured and single-turn paradigm, are insufficient to tackle the full complexity of real-world problems: *this limitation motivates the transition of agents capable of engaging in dynamic, multi-turn interactions with external tools and environments (Wang et al., 2025c; Mai et al., 2025; Feng et al., 2025b).*

However, the transition to multi-turn tool-use settings introduces several key challenges:

- **Data scarcity** (ℂ1)**:** High-quality multi-turn tool-use datasets are exceedingly scarce due to the labor-intensive nature of human annotation and validation. For instance, BFCL V3 (Patil et al., 2025b) multi-turn dataset contains only 800 samples, severely limiting the effectiveness of traditional data-driven approaches.
- **Complex environment** (ℂ2)**:** Multi-turn tool-use scenarios require agents to navigate complex, interconnected tool ecosystems spanning multiple domains. Like BFCL V3, the benchmark spans 8 different domains with 84 distinct tools, requiring cross-domain API calls and sophisticated tool orchestration.

---

*Equal contribution. Work was done during internship in Ant Group.
†Corresponding author.

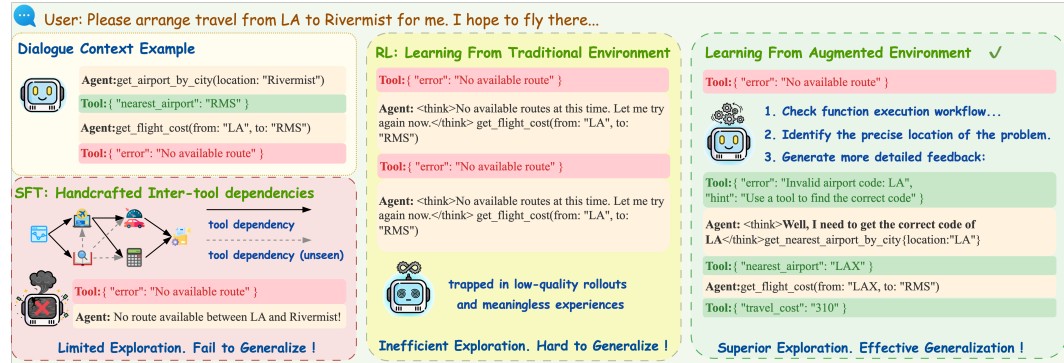

Figure 1: **Limitations of Existing Paradigms and the ENVIRONMENT TUNING Advantage.** This figure contrasts three agent training approaches on a travel planning task. **(Left)** Supervised Fine-Tuning (SFT) on static trajectories struggles with generalization. **(Center)** Reinforcement Learning (RL) in a traditional environment provides only sparse, uninformative feedback. **(Right)** Our approach uses an **augmented environment** that provides actionable, fine-grained feedback upon failure.

- **Long interaction chain** ($\mathbb{C}3$)**:** Success in multi-turn scenarios demands consistent performance across all interaction turns, where any single failure leads to complete task failure. Each test sample involves multiple user queries, where success requires passing all checks in every turn.

While constructing synthetic trajectories for Supervised Fine-Tuning (SFT) is a dominant strategy (Prabhakar et al., 2025; Liu et al., 2024; Yin et al., 2025) to mitigate data scarcity ($\mathbb{C}1$), it suffers from a critical flaw: agents trained on these static, synthetic traces often fail to generalize to real-world scenarios, a limitation we demonstrate empirically in Section 4.2.

Reinforcement Learning (RL) (Shao et al., 2024; Yu et al., 2025; Hu et al., 2025) offers a natural alternative, promising to improve generalization through online interaction and exploration (Chu et al., 2025; Shenfeld et al., 2025). However, this approach is plagued by its own severe challenges. The complexity of the environment ($\mathbb{C}2$) creates a critical "cold-start" problem: an agent that is not yet proficient cannot explore the vast action space effectively, becoming trapped in cycles of low-quality rollouts and failing to generate the meaningful experiences required for improvement. Furthermore, even if an agent overcomes this initial hurdle, the long interaction chains ($\mathbb{C}3$) inherent to these tasks make the training process notoriously unstable and prone to performance collapse (Wang et al., 2025c; Xue et al., 2025).

This collectively leads to our central research question:

> *How can we train a high-quality agent for complex, multi-turn tool use under extreme **data scarcity**, ensuring both **generalization** and **stability**?*

Addressing the critical intersection of data scarcity, generalization, and learning stability requires a paradigm shift in agent training. In response, we introduce ENVIRONMENT TUNING, a novel framework designed to cultivate both generalization and stability from extremely scarce data. At its core, ENVIRONMENT TUNING orchestrates learning via three complementary principles: (1) a *Structured Curriculum* that guides the agent from simple to complex tasks to build skills progressively; (2) *Actionable Environment Augmentation* that provides corrective hints upon failure, turning dead-end explorations into rich learning signals; and (3) *Fine-Grained Progress Rewards* that replace sparse, binary outcomes with a continuous measure of task completion, providing a dense and informative learning signal.

By combining this deliberate curriculum with enriched feedback and dense rewards, ENVIRONMENT TUNING enables an agent to acquire sophisticated, multi-step behaviors from scratch, demonstrating that robust learning is achievable even in the complete absence of expert demonstrations. **The key contributions of this work are as follows:**

- **A novel learning paradigm for data-scarce environments.** We propose ENVIRONMENT TUNING, which enables agents to learn multi-turn tool-use capabilities directly from problem instances without expert demonstrations, shifting from trajectory-based imitation to environment-based exploration.

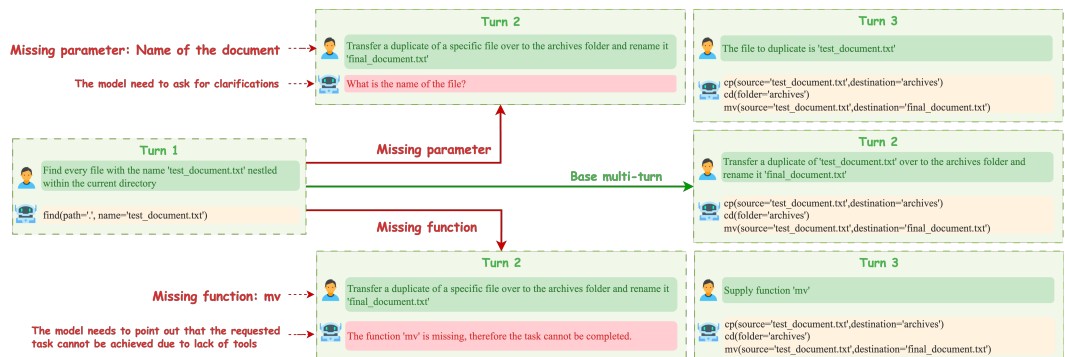

Figure 2: **An illustration of multi-turn tool-use scenarios**, adapted from an official example in the BFCL V3 Blog (Patil et al., 2025a). All three tracks start from the same initial user request. The *Base multi-turn* track (center) shows a successful execution path. The *Missing parameter* track (top) illustrates a scenario where the agent must handle ambiguity by asking for clarification. The *Missing function* track (bottom) shows a case where the agent needs to recognize that a required tool is unavailable. These scenarios highlight the diverse reasoning capabilities our curriculum is designed to address.

- **A practical curriculum with environment engineering.** We develop a four-stage curriculum leveraging actionable environment augmentation and fine-grained progress rewards to transform sparse feedback into rich learning signals for effective exploration.
- **Empirical validation in extreme data scarcity.** With only 400 training samples, our method proves effective for both base and SFT-tuned models. It lifts a base model like Qwen2.5-7B from near-zero to strong in-distribution performance, and also boosts SFT-tuned models such as watt-tool-8B to 54.25%, surpassing most proprietary models. Notably, it nearly doubles ToolACE-2's out-of-distribution score on ACEBench (8.5% to 15.0%), showing that ENVIRONMENT TUNING fosters robust generalization where SFT often fails.

## 2 RELATED WORK

**Tool-integrated reasoning.** A prominent line of research focuses on augmenting LLMs with external tools, a paradigm often termed Tool-Integrated Reasoning (TIR). Many works have leveraged RL as an alternative to SFT for teaching agents strategic tool invocation (Li et al., 2025; Feng et al., 2025a). However, direct application of trajectory-level algorithms (Shao et al., 2024; Yu et al., 2025) often leads to training instability and performance collapse (Wang et al., 2025c; Mai et al., 2025; Xue et al., 2025). To address this challenge, recent works have proposed sophisticated mechanisms like fine-grained credit assignment (Feng et al., 2025b), entropy-guided exploration (Dong et al., 2025), and trajectory filtering to prevent gradient explosion (Xue et al., 2025). While effective in their respective domains, such as computational reasoning (Li et al., 2025; Singh et al., 2025) or open-domain web search (Jin et al., 2025a; Zheng et al., 2025), these methods are typically evaluated in settings where the primary challenge is to master a single tool or a small, homogeneous set of tools. Our work, in contrast, addresses the distinct challenge of orchestrating a large and diverse toolset to complete complex, stateful tasks that unfold over multiple interaction turns.

**Multi-turn tool orchestration.** Distinct from tool-augmented reasoning, another major challenge involves enabling agents to orchestrate a large set of diverse APIs to accomplish complex, stateful tasks over multiple turns. Benchmarks like BFCL (Patil et al., 2025b) and ACEBench (Chen et al., 2025a) have emerged to evaluate these sophisticated capabilities. However, as these benchmarks are derived from realistic scenarios, they feature high-quality but scarce human-annotated data, posing a significant training challenge.

To address the aforementioned **data scarcity** issue, two primary strategies have been explored. The dominant approach involves large-scale synthetic data generation for SFT, where recent works (Prabhakar et al., 2025; Liu et al., 2024; Yin et al., 2025; Zhang et al., 2024) focus on creating vast corpora of tool-use trajectories. An alternative strategy explores applying online RL directly. Works such as ReCall (Chen et al., 2025b) and ARTIST (Singh et al., 2025) attempt to improve policies through direct environment interaction. However, these online RL approaches have so far yielded only modest gains on benchmarks like BFCL, highlighting the difficulty of effective exploration in a **complex environment** and motivating the need for more efficient learning paradigms.

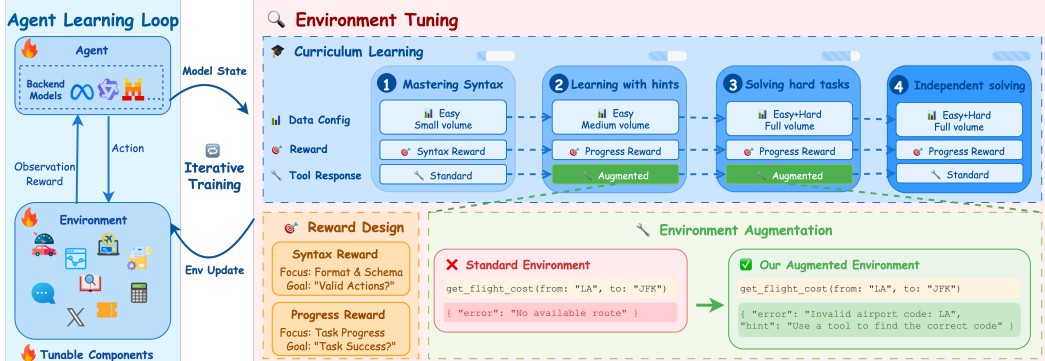

Figure 3: **An overview of ENVIRONMENT TUNING.** Our core innovation is the ENVIRONMENT TUNING module, which implements a four-stage curriculum. It dynamically configures the reward function, environment feedback (Standard vs. Augmented), and data split for the *Agent Learning Loop*. This staged approach transforms ambiguous errors into actionable lessons (highlighted in the *Environment Augmentation in Action* panel), enabling efficient and stable learning from limited data.

## 3 ENVIRONMENT TUNING

To overcome the challenge of learning in complex, data-scarce environments, we propose ENVIRON-MENT TUNING, a framework that orchestrates training through three complementary mechanisms (see Figure 3). It combines (1) a *Structured Curriculum* (Section 3.2) for progressive skill acquisition; (2) *Actionable Environment Augmentation* (Section 3.3) to turn failures into rich learning signals via corrective hints; and (3) a *Fine-Grained Progress Reward* (Section 3.4) to provide dense feedback that overcomes the limitations of sparse signals. These pillars collectively create a manageable learning path for stable and data-efficient agent training.

### 3.1 PRELIMINARY AND CHALLENGES

This work tackles ***multi-turn tool use***: a challenging task where an agent must achieve a complex goal through a series of interactions with external tools. Unlike single-step problems, this process is dynamic and demands sophisticated reasoning, as a single task can unfold in numerous ways (Figure 2). The agent must navigate an interactive loop of understanding user requests, executing tool calls, and generating responses until the overall objective is complete. We mathematically formalize this problem as a Partially Observable Markov Decision Process (POMDP) in Section B.

**On the challenge of multi-turn tool use.** Training an RL agent for multi-turn tool use is non-trivial: an agent must simultaneously master low-level syntax skills (e.g., precise tool-call formatting that the environment can parse) and high-level reasoning abilities (e.g., multi-step planning across dependent subtasks). In complex interactive environments, base models lacking these skills exhibit a diverse range of failure modes, including void incorrect parameter filling, calls to non-existent tools, and irreversible environment corruption.

As noted by Xue et al. (2025), RL optimization on such noisy trajectories is extremely sensitive to these error patterns: if they persist in the rollouts, training instability and gradient explosion are likely. Our numerical experiments also confirm this fragility: *when fine-tuning Qwen2.5-7B-Instruct directly in a single-stage RL setup with 400 training instances, training collapsed within 70 steps, yielding a mere 10% improvement in success rate*, as detailed and visualized in Appendix Section F.2.

### 3.2 STRUCTURED CURRICULUM RL TRAINING

The aforementioned observations suggest that naively optimizing for full task success from the start is ineffective in long-horizon, sparse-reward settings. We therefore propose a structured curriculum that gradually increases objective complexity. This four-stage curriculum allows the agent to progress from mastering foundational skills to handling the full complexity of multi-turn tool use, while maintaining training stability.

**Stage 1: mastering syntactic and schematic correctness.** The goal of this initial stage is to train the agent to produce well-formed outputs with valid tool calls, forming a reliable foundation for all subsequent learning. Before an agent can reason effectively, it must first "speak the language" of the environment; otherwise, learning is confounded by penalties for both poor strategy and invalid formatting.

To isolate this foundational skill, we deconstruct the agent's actions ($a_t^{\text{tool}}$ and $a_t^{\text{answer}}$) and design a reward function focused exclusively on their structural integrity. As illustrated in Figure 3, we use several task-agnostic, turn-by-turn counters:

- $C_{correct}$: the number of tool calls with a correct tool name and valid arguments;
- $C_{error}$: the number of calls with a correct tool name but invalid arguments;
- $C_{format}$: the number of turns violating the required XML-like format.

These counters are combined into a *Format Reward* $R_{\text{format}}$ and a *Tool Call Reward* $R_{\text{tool}}$, defined as

$$R_{\text{format}} = (N - C_{\text{format}})/N \,, \qquad R_{\text{tool}} = C_{\text{correct}}/(C_{\text{correct}} + C_{\text{error}}) \,, \tag{1}$$

where $N$ is the total number of turns in the episode. The final reward for this stage, $R_{\text{Stage1}}$, combines these components and is gated by an indicator $I_{\text{tool}}$ that is 1 if the agent attempts at least one tool call and 0 otherwise, encouraging active tool use:

$$R_{\text{Stage1}} = I_{\text{tool}} \cdot (R_{\text{format}} + R_{\text{tool}}) \,. \tag{2}$$

As we demonstrate in our analysis of training dynamics (Section F.1), this specialized reward allows the agent to rapidly master the required syntax. This proves that dedicating a stage to this foundational skill is a crucial prerequisite for efficient learning in more complex, task-oriented stages.

**Stage 2: basic learning with augmented feedback.** With the agent now proficient in syntax, the curriculum transitions to learning task-oriented reasoning. This stage utilizes the full `Base` split from BFCL, focusing on foundational multi-turn capabilities. To accelerate learning, we introduce two critical components to allow the agent to efficiently learn core reasoning skills in a guided manner.

1. Progress Reward (c.f. Section 3.4): we replace the task-agnostic reward with a Progress Reward ($R_P$), a fine-grained measure of task completion that provides a more informative signal than sparse binary outcomes (detailed in Section 3.4).
2. Actionable Environment Augmentation (c.f. Section 3.3): we employ such augmentation to enable the environment to provide detailed, corrective hints upon failure instead of ambiguous error messages, turning failed explorations into valuable learning opportunities.

**Stage 3: advanced learning on complex scenarios.** Building on the foundational skills from the previous stage, the agent is now exposed to the full spectrum of challenges. We introduce the complete training dataset, incorporating samples from the `Missing Parameters`, `Missing Functions`, and `Long-Context` splits. The objective is to train the agent to handle ambiguity, recognize functional gaps, and perform information retrieval from noisy contexts. The training setup remains consistent with Stage 2, continuing to leverage both the Progress Reward and Actionable Environment Augmentation to help the agent navigate these more complex problem spaces.

**Stage 4: alignment with the evaluation environment.** The final stage is designed to align the agent with the true evaluation conditions. While still training on the full dataset with the Progress Reward, we now disable the Actionable Environment Augmentation. This forces the agent to generalize its learned policies, relying on its internal reasoning capabilities to handle uninformative or standard error messages, just as it would during final evaluation. This step is crucial for ensuring that the agent's performance is robust in OOD scenarios and not overly dependent on the training scaffolds provided in earlier stages.

**Remark 3.1** (Checkpoint selection for stage transitions). *A key element of our curriculum is determining when to transit to the next stage. In our multi-turn tool-use setting, we adopt a **validation- and stability-based stage transition rule**: we advance to the next stage only when the validation accuracy has plateaued and its gradient norm is stable.*

*This joint condition, further discussed in Section F.1, serves two purposes. First, the converged validation performance ensures the agent has mastered the current stage's skills. Second, the stable optimization dynamics confirm it is ready for more complex tasks, mitigating the risk of gradient explosions common in long-horizon RL (Xue et al., 2025). This rule is crucial for maintaining training stability and maximizing the curriculum's effectiveness.*

### 3.3 ACTIONABLE ENVIRONMENT AUGMENTATION

In multi-turn tool-use tasks, an agent often needs to execute complex chains of function calls where the output of one tool becomes the input to another. However, standard training environments typically

return cryptic or overly generic error messages (like simple error codes or "not found" responses) that neither identify the root cause of failure nor indicate a viable next action. This lack of actionable guidance severely limits the agent's ability to discover dependencies between tools and to understand each tool's usage constraints, forcing it to rely on inefficient trial-and-error exploration, which leads to poor performance (as validated by the ablation in Section 4.3 that isolates the effect of environment augmentation). To address these issues, we design Actionable Environment Augmentation, which modifies the environment's feedback to provide pedagogical hints that directly inform the agent about dependency relationships and operational rules, turning failed trajectories into constructive learning opportunities.

**Discovering inter-tool dependencies via guided discovery.** Our first goal is to empower the agent to discover and resolve inter-tool dependencies on its own. While prior work often relies on pre-constructed dependency graphs to explicitly teach agents sequential tool-call patterns (Prabhakar et al., 2025; Liu et al., 2024), our method embeds these dependencies within the environment's feedback. This encourages the agent to learn the underlying logic through interaction and inference rather than memorization, leading to better generalization in unseen scenarios.

For example, in the BFCL Travel API task, an agent might incorrectly try to book a flight without first finding the correct airport code.

- *Standard feedback*: A vague message like "`No available route`" gives the agent no clue about the root cause of the failure.
- *Augmented feedback*: Our environment provides a precise hint: "`Invalid airport code[s]:...`". This message suggests that another tool is needed to find the correct airport code first, effectively prompting the agent to discover the dependency through interaction.

**Revealing internal tool constraints with pedagogical hints.** Our second objective is to provide actionable hints that reveal a tool's specific internal rules and constraints. This moves beyond the simple diagnostic errors returned by standard code interpreters (e.g., "`FileNotFoundError`") toward providing pedagogical feedback that explains why an action failed. By revealing the "rules of the game" to the agent, we dramatically prune the exploration space by invalidating entire classes of incorrect attempts.

For instance, an agent's pretrained knowledge might suggest using full paths with the "`rm`" command. However, the BFCL File System environment may not support this.

- *Standard feedback*: A generic "`FileNotFoundError`" would be misleading, as the file might actually exist.
- *Augmented feedback*: Our environment returns an explicit rule: "`Paths are not allowed. Specify only file/directory name...`". This hint directly corrects the agent's misunderstanding of the tool's protocol within this specific environment.

This LLM-driven workflow for generating augmentations is detailed in Section C. As illustrated in the file system and multi-API travel case studies (Sections H.1 and H.2), our augmented feedback transforms failed turns into explicit, actionable guidance. This enables agents to diagnose root causes accurately and discover corrective strategies that would be infeasible under baseline environments with ambiguous signals.

### 3.4 FINE-GRAINED PROGRESS REWARD

A key challenge in training long-horizon, multi-turn tool-use agents is *reward sparsity*: a single binary signal at the end of a long trajectory provides insufficient guidance for effective learning (Feng et al., 2025b).

To overcome these challenges, we introduce a fine-grained *Progress Reward* ($R_P$) that provides a denser, turn-by-turn learning signal. Instead of rewarding individual tokens, we evaluate success at the end of each turn based on two criteria: the correctness of the resulting *environment state* and the *execution result* of the agent's chosen action. This allows us to distinguish "nearly correct" from "completely wrong" trajectories.

Formally, for each turn $t$ in an episode of length $T$, we define binary scores for the state evaluation ($r_t^{\text{state}}$) and execution result evaluation ($r_t^{\text{exec}}$). A turn is successful only if both are correct, with its reward being their product: $r_t = r_t^{\text{state}} \cdot r_t^{\text{exec}}$. The total Progress Reward $R_P$ is then the average success rate across all turns: $R_P = \frac{1}{T} \sum_{t=1}^{T} r_t = \frac{1}{T} \sum_{t=1}^{T} (r_t^{\text{state}} \cdot r_t^{\text{exec}})$. This formulation provides a rich, informative signal throughout the episode, enabling the agent to learn efficiently from partially

Table 1: **Main results on the BFCL V3 multi-turn benchmark.** Our method, ENVIRONMENT TUNING, significantly boosts the performance of all base models, achieving competitive results against proprietary models and outperforming several strong baselines using only 400 training samples.

| Model | BFCL V3 Multi Turn | | | | |
|---|---|---|---|---|---|
| | Average (%) | Base (%) | Miss Func (%) | Miss Param (%) | Long Context (%) |
| Claude Sonnet 4 | 57.00 | 63.00 | 58.00 | 51.00 | 56.00 |
| GPT-4o | 51.00 | 59.00 | 54.00 | 41.00 | 50.00 |
| Gemini 2.5 Pro | 28.75 | 32.00 | 29.00 | 22.00 | 32.00 |
| o3 | 49.25 | 47.00 | 55.00 | 47.00 | 48.00 |
| Arch-Agent-7B | 42.05 | 47.15 | 53.75 | 34.20 | 33.10 |
| xLAM-2-8b-fc-r | 70.50 | 77.85 | 69.15 | 65.80 | 69.20 |
| BitAgent-8B | 36.99 | 47.85 | 33.20 | 26.15 | 40.75 |
| Qwen2.5-7B-Instruct | 7.00 | 9.33 | 9.33 | 6.33 | 3.00 |
| + ToolRL | 18.00 (+11.00) | 23.00 (+13.67) | 26.00 (+16.67) | 16.00 (+9.67) | 7.00 (+4.00) |
| + ENVIRONMENT TUNING | 36.92 (+29.92) | 50.33 (+41.00) | 40.33 (+31.00) | 29.33 (+23.00) | 27.67 (+24.67) |
| Llama-3.1-8B-Instruct | 5.48 | 6.15 | 6.80 | 3.20 | 5.75 |
| + ToolRL | 11.25 (+5.77) | 9.00 (+2.85) | 10.00 (+3.20) | 9.00 (+5.80) | 17.00 (+11.25) |
| + ENVIRONMENT TUNING | 28.25 (+22.77) | 28.20 (+22.05) | 25.85 (+19.05) | 22.15 (+18.95) | 36.80 (+31.05) |
| ToolACE-2-Llama-3.1-8B | 37.99 | 48.85 | 34.15 | 25.20 | 43.75 |
| + ToolRL | 33.75 (-4.24) | 46.00 (-2.85) | 29.00 (-5.15) | 22.00 (-3.20) | 38.00 (-5.75) |
| + ENVIRONMENT TUNING | 47.18 (+9.19) | 55.20 (+6.35) | 38.15 (+4.00) | 38.20 (+13.00) | 57.15 (+13.40) |
| watt-tool-8B | 35.74 | 45.85 | 33.15 | 25.20 | 38.75 |
| + ToolRL | 42.00 (+6.26) | 55.00 (+9.15) | 35.00 (+1.85) | 31.00 (+5.80) | 47.00 (+8.25) |
| + ENVIRONMENT TUNING | 54.34 (+18.50) | 64.15 (+18.30) | 48.15 (+15.00) | 40.20 (+15.00) | 64.85 (+26.10) |

successful attempts while still having the freedom to discover novel problem-solving strategies. A detailed breakdown of the evaluation criteria is provided in Section D.

# 4 EXPERIMENT

## 4.1 EXPERIMENT SETTINGS

**Benchmark.** Our primary evaluations are conducted on the multi-turn subset of the Berkeley Function-Calling Leaderboard (BFCL) V3 (Patil et al., 2025b). This benchmark comprises a total of 800 samples, divided equally into four challenging splits: `Base`, `Missing Functions`, `Missing Parameters`, and `Long-Context`. More specifically, we construct a training set of 400 samples by selecting 100 from each split. The remaining 400 samples serve as our held-in test set to evaluate in-distribution performance. For out-of-distribution (OOD) evaluation, we use the BFCL V4 `web search` and `memory` tracks (Patil et al., 2025b), $\tau^2$-bench (Barres et al., 2025) and ACEBench Agent split (Chen et al., 2025a) as our held-out test set. Notably, our training set (BFCL V3) is confined to closed-domain API calls, while our OOD benchmarks introduce entirely distinct domains and capabilities. This clear separation ensures a rigorous evaluation of generalization. A detailed description of these benchmarks and their evaluation methodologies is available in Section E.1.

**Models.** To comprehensively evaluate our approach, we categorize the models used in our experiments into three groups.

- **Base models:** These are the open-source models upon which we apply ENVIRONMENT TUNING method. We select `Qwen2.5-7B-Instruct` (Qwen et al., 2025) and `Llama-3.1-8B-Instruct` (Grattafiori et al., 2024) as our primary base models. As successfully applying reinforcement learning to Llama-based models has proven difficult (Zeng et al., 2025; Wang et al., 2025b; Gandhi et al., 2025), we also include two SFT-tuned versions of Llama-3.1-8B-Instruct: `ToolACE-2-Llama-3.1-8B` (Liu et al., 2024) and `watt-tool-8B`[1]. Using these SFT-tuned models as base models allows us to demonstrate the general applicability of ENVIRONMENT TUNING on models that are already strong in tool use.
- **Open-source baselines:** To contrast our environment-centric RL approach with prevailing data-driven methods, we compare against four state-of-the-art SFT-tuned models: `Arch-Agent-7B`[2], `xLAM-2-8b-fc-r` (Prabhakar et al., 2025), and `BitAgent-8B`[3]. As discussed in Section 2,

---

[1] https://huggingface.co/watt-ai/watt-tool-8B
[2] https://huggingface.co/katanemo/Arch-Agent-7B
[3] https://huggingface.co/BitAgent/BitAgent-8B

Table 2: **OOD Generalization performance on BFCL V4, $\tau^2$-bench, and ACEBench Agent benchmarks**. All results are compared against the **Llama-3.1-8B-Instruct** base model. Blue text indicates a performance improvement over the base model, while orange text indicates a performance degradation. Models trained with ENVIRONMENT TUNING (rows in blue) show consistent improvements, whereas SFT baselines (rows in red) often underperform the base model on these OOD tasks. Scores for xLAM on the Retail and Airline domains are grayed out as it was trained on the original $\tau$-bench, making them invalid for OOD evaluation.

| Model | BFCL V4 | | | $\tau^2$-bench | | | | ACEBench Agent | | |
|---|---|---|---|---|---|---|---|---|---|---|
| | Avg. (%) | Web Search (%) | Memory (%) | Avg. (%) | Retail (%) | Airline (%) | Telecom (%) | Avg. (%) | Multi-turn (%) | Multi-step (%) |
| xLAM-2-8b-fc-r | 9.17 | 5.00 | 13.33 | 36.67 | 57.50 | 40.00 | 12.50 | 1.65 | 0.00 | 3.33 |
| BitAgent-8B | 7.41 | 4.50 | 10.32 | 10.00 | 7.50 | 15.00 | 7.50 | 5.00 | 10.00 | 0.00 |
| Llama-3.1-8B-Instruct | 8.46 | 1.00 | 15.91 | 20.00 | 2.50 | 30.00 | 27.50 | 1.65 | 0.00 | 3.33 |
| + ToolRL | 11.52 | 11.00 | 12.04 | 21.67 | 10.00 | 25.00 | 30.00 | 1.65 | 0.00 | 3.33 |
| + ENVIRONMENT TUNING | 16.53 | 15.00 | 18.06 | 21.67 | 5.00 | 30.00 | 30.00 | 4.17 | 5.00 | 3.33 |
| ToolACE-2-Llama-3.1-8B | 15.90 | 9.00 | 22.80 | 10.83 | 10.00 | 15.00 | 7.50 | 8.34 | 10.00 | 6.67 |
| + ToolRL | 18.91 | 9.00 | 28.82 | 14.17 | 5.00 | 30.00 | 7.50 | 6.65 | 3.30 | 10.00 |
| + ENVIRONMENT TUNING | 16.79 | 14.00 | 19.57 | 20.83 | 20.00 | 30.00 | 12.50 | 15.00 | 10.00 | 20.00 |
| watt-tool-8B | 8.67 | 4.00 | 13.33 | 13.33 | 15.00 | 20.00 | 5.00 | 2.50 | 5.00 | 0.00 |
| + ToolRL | 13.12 | 1.50 | 24.73 | 11.67 | 12.50 | 15.00 | 7.50 | 9.15 | 3.30 | 15.00 |
| + ENVIRONMENT TUNING | 13.68 | 8.00 | 19.35 | 15.83 | 20.00 | 20.00 | 7.50 | 7.50 | 0.00 | 15.00 |

these models are representative of the dominant data synthesis paradigm to handle data scarcity issue ($\mathbb{C}1$).

- **Proprietary models:** To benchmark against the absolute state-of-the-art, we include leading proprietary models such as `Claude Sonnet 4` (Anthropic, 2025), `GPT-4o` (Hurst et al., 2024), `Gemini 2.5 Pro` (Comanici et al., 2025), and `o3` (OpenAI, 2025b). These models serve as a reference for top-tier performance.

**Agent training.** For agent training, we employ an adapted version of the Group-Relative Policy Optimization (GRPO) algorithm (Shao et al., 2024), enhanced with a decoupled clipping mechanism and a KL-divergence penalty to ensure stable and effective exploration; full implementation details are provided in Section E.2. As a baseline, we also implement ToolRL (Qian et al., 2025), which applies the GRPO algorithm directly with a reward function of $R_{\text{format}} + R_{\text{correct}}$, where $R_{\text{correct}}$ measures the correctness of tool calls.

## 4.2 MAIN RESULTS

**Results on multi-turn tool use.** Our method demonstrates substantial effectiveness in the in-distribution, multi-turn tool use scenarios presented in BFCL V3. As shown in Table 1, ENVIRONMENT TUNING yields significant performance gains across all base models.

- **Significant performance uplift from scratch.** When applied directly to base models, ENVIRONMENT TUNING proves to be a highly effective training paradigm that consistently works across different model architectures. For instance, it boosts Qwen2.5-7B-Instruct's score from $7.00\%$ to $36.92\%$, surpassing two strong baselines (BitAgent-8B and Arch-Agent-7B) and the proprietary Gemini 2.5 Pro model.
- **Effective enhancement of SFT-tuned models.** Our method also serves as a powerful online refinement tool for models that have already undergone SFT. On ToolACE-2-Llama-3.1-8B, a model built upon the RL-challenging Llama architecture, ENVIRONMENT TUNING still provides a significant improvement ($9.19\%$). This elevates its performance to $47.18\%$, surpassing the proprietary Gemini 2.5 Pro model. Similarly, on watt-tool-8B, ENVIRONMENT TUNING achieves an impressive $18.50\%$ improvement, boosting performance from $35.74\%$ to $54.34\%$, which exceeds most proprietary models including o3 and GPT-4o.
- **Superiority over direct RL.** Our method consistently and substantially outperforms the ToolRL baseline. While ToolRL alone yields only modest gains and can be unstable on SFT-tuned models (e.g., degrading ToolACE-2-Llama-3.1-8B's performance by $-4.24\%$), ENVIRONMENT TUNING delivers much larger improvements across the board (e.g., $+29.92\%$ on Qwen2.5-7B-Instruct and $+9.19\%$ on ToolACE-2-Llama-3.1-8B), underscoring the necessity of our curriculum and environment augmentations for effective learning.

**Results on OOD agentic tasks.** One strength of ENVIRONMENT TUNING lies in its ability to foster robust generalization, which we evaluate on the OOD benchmarks from BFCL V4, $\tau^2$-bench, and ACEBench Agent (Table 2). The results reveal a clear distinction between ENVIRONMENT TUNING and supervised fine-tuning on trajectories.

- **Superior generalization over SFT baselines.** The limitations of overfitting to static datasets become evident here. Both SFT baselines exhibit a dramatic performance collapse on the OOD Web Search task, with even the top-performer, xLAM-2 ($70.50\%$ on BFCL V3), dropping to

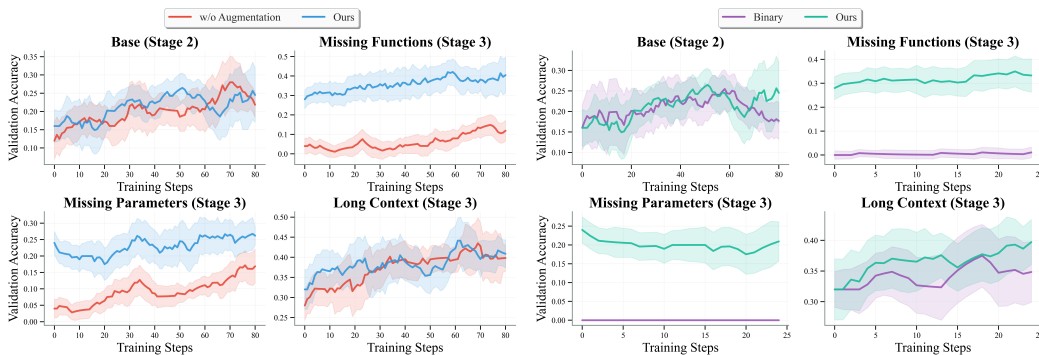

(a) Ablation study for environment augmentation.  (b) Ablation study for progress reward.

Figure 4: **Training dynamics comparison for ENVIRONMENT TUNING on Qwen2.5-7B-Instruct.** (a) The effect of Actionable Environment Augmentation on learning stability and performance across different data splits. (b) The impact of fine-grained Progress Reward versus binary reward on training effectiveness, showing the critical role of dense reward signals in complex multi-turn scenarios.

just $5.00\%$. This pattern persists across diverse OOD benchmarks, which underscores a critical weakness in training solely on synthetic trajectories.

- **ENVIRONMENT TUNING fosters robust generalization.** In contrast to the SFT baselines, our method demonstrates superior generalization by training agents through direct environmental interaction. For instance, ENVIRONMENT TUNING transforms Llama-3.1-8B-Instruct, which performs poorly on the Web Search task ($1.00\%$), into a strong performer at $15.00\%$, showcasing its ability to teach general problem-solving principles rather than dataset-specific patterns.
- **Enhances generalization of already proficient models.** Our method can also patch the generalization gaps left by SFT. The base ToolACE-2 model already shows decent OOD performance ($9.00\%$ on Web Search), far exceeding other baselines. Yet, ENVIRONMENT TUNING further enhances its capability, increasing its score to $14.00\%$, demonstrating that interactive learning is crucial for building truly adaptable agents. This enhancement extends to ACEBench Agent, where ENVIRONMENT TUNING boosts ToolACE-2's average score from $8.34\%$ to $15.00\%$.
- **Advantage over direct RL on complex OOD tasks.** Our framework demonstrates stronger and more stable generalization than the ToolRL baseline, particularly on complex benchmarks like ACEBench and $\tau^2$-bench. For instance, on ACEBench Agent, ENVIRONMENT TUNING lifts ToolACE-2-Llama-3.1-8B's score from $8.34\%$ to $15.00\%$, whereas direct RL causes a performance drop to $6.65\%$. This highlights that our curriculum and augmentations are key to fostering robust, generalizable capabilities not achieved by direct RL alone.

## 4.3 ABLATION STUDY

**Effect of structured curriculum.** To isolate the impact of our four-stage curriculum, we conduct an ablation study comparing our full method against two baselines on the Qwen2.5-7B-Instruct model: (1) the base pre-trained model, and (2) a direct GRPO training baseline. For this baseline, the agent is trained on the full 400-sample dataset from the start using a combined reward function of $0.9 \cdot R_P + 0.1 \cdot R_{\text{format}}$, without changing all other hyperparameters. As shown in Table 3, directly applying RL yields minimal gains, highlighting the "cold-start" problem where the agent fails to learn

Table 3: **Effectiveness of the structured curriculum on Qwen2.5-7B.** Performance comparison across different training stages, showing how each stage contributes to overall improvement. Abbreviations: M. Func = Missing Functions, M. Param = Missing Parameters, L. Ctxt = Long-Context.

| Training Stage | BFCL V3 Multi Turn | | | | |
|---|---|---|---|---|---|
| | Avg. | Base | M. Func | M. Param | L. Ctxt |
| Qwen2.5-7B-Instruct | 7.00 | 9.33 | 9.33 | 6.33 | 3.00 |
| + GRPO | 17.42 | 20.00 | 24.67 | 14.67 | 10.33 |
| + Stage 1 | 15.50 | 19.00 | 22.33 | 9.33 | 11.33 |
| + Stage 2 | 25.83 | 32.00 | 33.67 | 20.00 | 17.67 |
| + Stage 3 | 32.00 | 44.67 | 34.33 | 25.33 | 23.67 |
| + Stage 4 (ours) | **36.92** | **50.33** | **40.33** | **29.33** | **27.67** |

effectively. In contrast, our curriculum provides a clear and steady improvement path. *Each stage brings varying degrees of improvement, ultimately boosting the final performance to 36.92%, a 19.50% increase over the direct GRPO baseline.*

**Effect of actionable environment augmentation.** To validate the contribution of our Actionable Environment Augmentation, we conduct an ablation study comparing training dynamics with and

without the augmented feedback mechanism. As illustrated in Figure 4a, the augmented environment provides crucial learning signals that enable the agent to navigate complex scenarios more effectively.

- The augmented environment consistently leads to more stable learning curves across all data splits, particularly in the challenging `Missing Parameters` and `Missing Functions` scenarios.
- *For the* `Missing Parameters` *and* `Missing Functions` *splits, Environment Augmentation brings substantial performance improvements of over 20%, demonstrating its critical role in enabling effective learning on these complex, ambiguous tasks.*

**Effect of fine-grained progress reward.** We also conduct an ablation study to verify the effectiveness of our fine-grained Progress Reward ($R_P$) against a standard binary reward. As shown in Figure 4b, the impact of the reward design varies significantly with task complexity:

- During Stage 2, which focuses on the `Base` data, the performance difference is subtle, suggesting that a simple binary signal is sufficient for foundational tasks.
- However, as the curriculum progresses to more complex splits in Stage 3, the necessity of a denser reward becomes starkly evident. *For the* `Missing Parameters` *and* `Missing Functions` *splits, the binary reward leads to complete training failure, with performance close to zero, as the sparse signal provides no incentive for the necessary exploratory actions.*
- Meanwhile, on the `Long Context` split, our Progress Reward leads to substantially more stable and effective learning.

## 5 CONCLUSION

In this work, we proposed ENVIRONMENT TUNING, a novel training paradigm for multi-turn, tool-augmented agents under extreme data scarcity. By shifting the focus from trajectory imitation to environment-driven exploration, and combining a structured curriculum, actionable environment augmentation, and fine-grained progress rewards, our method enables agents to learn stably and generalize from only 400 problem instances without expert demonstrations. Experiments on BFCL show that ENVIRONMENT TUNING not only yields substantial in-distribution improvements but also significantly enhances out-of-distribution generalization, outperforming several strong supervised baselines. We believe this environment-centric approach offers a promising direction for developing robust, adaptable agents in realistic, resource-limited settings. Our work also opens several exciting avenues for future research, including the development of automated mechanisms for curriculum and feedback generation and the extension of ENVIRONMENT TUNING to more complex, multi-modal agentic scenarios.

## 6 ACKNOWLEDGMENT

This work was supported in part by the NSFC under No. 62576285, the Ant Group Research Collaboration Project, Research Center for Industries of the Future (RCIF) at Westlake University, and Westlake Education Foundation.

## 7 ETHICS STATEMENT

This work uses only publicly available datasets and open-source models, involves no human subjects or sensitive data, and complies with the ICLR Code of Ethics.

## 8 REPRODUCIBILITY STATEMENT

To support the reproducibility of our findings, we commit to releasing our complete source code, including implementations of the ENVIRONMENT TUNING framework and all training and evaluation scripts, as supplementary material. The core principles of our method are presented in Section 3. Detailed specifications essential for replication are provided throughout the appendix: the formal problem definition is in Section B, the progress reward mechanism is in Section D, and the full prompt templates used for training are in Section G. Concrete examples of our actionable environment augmentation are available in the case studies in Section H. A comprehensive description of our experimental setup, covering the datasets (Section 4.1 and Section E.1), models (Section 4.1), and the RL training algorithm with all hyperparameters (Section E.2), is provided. Furthermore, Section F contains supplementary experiments and analyses on training dynamics and stability to aid in the successful replication of our results.

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

CONTENTS

## A   USE OF LLMS

This paper only uses LLMs for polishing.

## B   SEQUENTIAL MULTI-TURN DECISION-MAKING MODEL.

We model the multi-turn tool-use task as a Partially Observable Markov Decision Process (POMDP) (Williams, 2007). An episode corresponds to a complete user task, which is composed of a sequence of pre-defined user requests, or "turns."

Let's denote the sequence of user requests as $q_1, q_2, ..., q_N$. The interaction begins with the agent receiving an initial observation $o_0$ containing the first request $q_1$ and the available tool documentation. The agent then engages in a series of steps to address this request. At each step $t$, the agent's policy $\pi_\theta(a_t|o_t)$ generates an action $a_t$ from a structured action space $\mathcal{A}$, which includes:

- **Tool Call** ($a_t^{\textbf{tool}}$):   A structured call to one or more tools to gather information (e.g., `<tool_call>...</tool_call>`). The environment executes the call and returns an observation containing the tool's output.
- **Final Answer** ($a_t^{\textbf{answer}}$):   A natural language response to the user (e.g., `<answer>...</answer>`).   This completes the current sub-task, after which the environment presents the next pre-defined user request.

The process for a single turn $i$ unfolds as a sub-trajectory of tool calls until the agent produces a final answer. Upon generating $a_t^{\text{answer}}$, the environment transitions by revealing the next request $q_{i+1}$ within the new observation $o_{t+1}$. This cycle repeats for all $N$ requests.

The entire episode, a trajectory $\tau = (o_0, a_0, o_1, a_1, ..., o_T)$, concludes only after the agent has provided a final answer for the last request, $q_N$. It is only at this point, after $T$ steps, that the agent receives a *sparse and terminal reward* $R_T \in \{0, 1\}$, indicating the overall success or failure of the entire multi-turn task. This delayed and binary feedback makes it extremely difficult for RL algorithms to perform effective credit assignment and exploration, a well-known challenge in long-horizon tasks (Wang et al., 2025c; Feng et al., 2025b).

The agent's objective is to learn the policy parameters $\theta$ that maximize the expected terminal reward:

$$J(\theta) = \mathbb{E}_{\tau \sim \pi_\theta, P}[R_T] \tag{3}$$

## C   IMPLEMENTATION OF ACTIONABLE ENVIRONMENT AUGMENTATION

A critical aspect of implementing our Actionable Environment Augmentation is ensuring its practicality and scalability. While crafting high-quality, pedagogical feedback might appear to require extensive manual engineering and deep domain expertise, we address this challenge with a semi-automated, LLM-driven workflow. We demonstrate that **leveraging the in-context learning capabilities of large language models, combined with a small set of few-shot examples, enables scalable and high-quality augmentation generation**.

Our implementation employs a two-agent workflow orchestrated by LLMs: (1) an **Augmentation Generator** that analyzes error trajectories and produces enhanced feedback, and (2) a **Constitutional Judge** that validates the augmented feedback to prevent solution leakage while ensuring actionability. This automated pipeline significantly reduces the manual burden of designing augmented feedback for each error case, making our method practical and generalizable to new domains.

Below, we present the core prompt templates that define these two agents, demonstrating how structured prompting with clear criteria and few-shot examples enables effective augmentation generation.

### C.1   AUGMENTATION GENERATOR PROMPT

The Augmentation Generator is responsible for determining whether an error message requires augmentation and, if so, producing enhanced feedback that is both actionable and pedagogical.

---
**Augmentation Generator System Prompt**

```
You are an expert at analyzing error messages in multi-turn
    tool-use scenarios.
```
---

```
Your task is to determine whether an error feedback needs
    augmentation to be more actionable and educational.

# Judgment Criteria

## 1. Need Augmentation (need_augmentation: true/false)
Does this error feedback need enhancement?
Consider:
- Is the error message vague or unclear?
- Does it lack actionable guidance?
- Does it hide critical dependency relationships?
- Does it fail to explain violated tool constraints?

## 2. Error Category (error_category)
What type of error is this?
Categories:
- dependency_missing: Missing prerequisite tool calls (need to
    call other tools first)
- parameter_constraint: Parameter constraint violation (incorrect
    parameter format or type)
- state_prerequisite: State precondition not met (incorrect
    environment state)
- format_mismatch: Format mismatch (incorrect input/output format)
- resource_unavailable: Resource unavailable (requested resource
    does not exist)

## 3. Augmentation Type (augmentation_type)
What type of enhancement is needed?
Types:
- clarify_constraint: Clarify constraints (explain tool usage
    rules)
- suggest_prerequisite: Suggest prerequisite steps (hint at what
    needs to be done first)
- explain_dependency: Explain dependencies (describe inter-tool
    dependencies)
- correct_format: Correct format errors (point out correct format
    requirements)

# Output Format
You must respond with a valid JSON object. Put your reasoning
    FIRST to encourage step-by-step thinking:
{
    "reasoning": "Your detailed analysis and explanation of why
        you made this judgment. Think step by step here.",
    "need_augmentation": true/false,
    "error_category": "category_name" or null,
    "augmentation_type": "type_name" or null,
    "augmented_feedback": "enhanced error message" or null
}

# Guidelines for Augmented Feedback
If augmentation is needed, generate feedback that:
1. **Clarifies the root cause** of the error
2. **Provides directional guidance** without revealing specific
    solutions
3. **Maintains exploration space** - don't prescribe exact steps
4. **Is pedagogical** - helps the agent learn the tool's
    constraints

Examples:
BAD (too specific): "You should call get_airport_code('San
    Francisco') first"
```

```
GOOD: "Invalid airport code. You may need to retrieve the correct
    code for the city first."

BAD (vague): "Error occurred"
GOOD: "File path format is incorrect. This tool only accepts
    filenames in the current directory, not full paths."
```

### Augmentation Generator User Prompt Template

```
# Task Context
Domain: $task_description
Question: $question
Trajectory: $trajectory

# Few-Shot Examples

## Example 1
Tool: $example_tool_name_1
Call: $example_tool_call_1
Original: $example_original_feedback_1
Augmented: $example_augmented_feedback_1
Reason: $example_reason_1

## Example 2
Tool: $example_tool_name_2
Call: $example_tool_call_2
Original: $example_original_feedback_2
Augmented: $example_augmented_feedback_2
Reason: $example_reason_2

[Additional examples as needed...]

# Current Case to Analyze
Tool Name: $tool_name
Tool Call: $tool_call
Original Feedback: $original_feedback

Please analyze whether this feedback needs augmentation and
    generate an improved version if needed.
```

## C.2 CONSTITUTIONAL JUDGE PROMPT

The Constitutional Judge ensures that augmented feedback maintains a balance between being helpful and preserving the agent's learning opportunity. It validates that the augmented feedback does not leak solutions or overly constrain exploration.

### Constitutional Judge System Prompt

```
You are a constitutional judge ensuring that augmented error
    feedback maintains a balance between being helpful and
    preserving the agent's learning opportunity.
Your task is to evaluate augmented feedback against strict
    criteria to prevent solution leakage and over-constraining
    exploration.

# Judgment Criteria
```

```
## 1. Solution Leakage (solution_leakage:
    true=violation/false=compliant)
Does the augmented feedback leak specific solutions?
Checks:
- Does it directly provide the correct function call?
- Does it explicitly specify concrete parameter values?
- Does it reveal the complete operation sequence?

Violation Examples:
- "You should call get_airport_code('San Francisco') first"
- "Use function A with parameter X=123, then call function B"

Good Examples:
- "Invalid airport code. You may need to retrieve the code first"
- "This operation requires certain prerequisites to be met"

## 2. Exploration Constraint (exploration_constraint:
    true=violation/false=compliant)
Does the augmented feedback overly constrain the exploration space?
Checks:
- Does it suggest only a single solution path?
- Does it exclude other reasonable attempts?
- Does it limit the agent's creative problem-solving?

Violation Examples:
- "You must use function A, then B, then C in this exact order"
- "The only way to solve this is to..."

Good Examples:
- "Consider checking if prerequisites are satisfied"
- "This error typically indicates a missing step earlier"

## 3. Actionability (actionability:
    true=qualified/false=unqualified)
Does the augmented feedback provide actionable guidance?
Checks:
- Does it clearly point out the root cause of the error?
- Does it provide improvement direction (not specific steps)?
- Does it help the agent understand tool usage rules?

## 4. Hint Quality (hint_quality: true=high/false=low)
Is the quality of the hint appropriate?
Checks:
- Is it specific enough to help diagnose the problem?
- Is it abstract enough to preserve exploration space?
- Does it follow pedagogical feedback principles?

# Output Format
You must respond with a valid JSON object. Put your reasoning
    FIRST to encourage step-by-step evaluation:
{
    "reasoning": "Your detailed evaluation and explanation.
        Analyze each criterion step by step.",
    "solution_leakage": true/false,  // true = violation (leaked
        solution)
    "exploration_constraint": true/false,  // true = violation
        (over-constrained)
    "actionability": true/false,  // true = qualified (provides
        actionable guidance)
    "hint_quality": true/false,  // true = high quality
    "overall_approved": true/false,  // true = passed review
    "suggestions": "suggestions for improvement if not approved"
        or null
```

```
    }

    # Approval Logic
    overall_approved = true IF AND ONLY IF:
    - solution_leakage == false (no leakage)
    - exploration_constraint == false (not over-constrained)
    - actionability == true (actionable)
    - hint_quality == true (qualified)
```

**Constitutional Judge User Prompt Template**

```
    # Feedback to Judge
    Question: $question
    Trajectory: $trajectory
    Tool Name: $tool_name
    Original Feedback: $original_feedback
    Augmented Feedback: $augmented_feedback

    # Ground Truth Reference (for your judgment only)
    $ground_truth_call_1
    $ground_truth_call_2
    [Additional ground truth calls as needed...]

    Please evaluate whether this augmented feedback passes all
        constitutional criteria.
```

### C.3 WORKFLOW AND PRACTICAL CONSIDERATIONS

The augmentation process is initiated by identifying challenging scenarios for the agent. During the initial stages of training, we use performance metrics to automatically flag questions where the agent repeatedly fails. The corresponding error-laden trajectories are collected, providing a rich dataset of common failure modes. These collected cases then serve as the input for our two-agent augmentation workflow.

The two agents work together in an iterative workflow:

1. The **Augmentation Generator** analyzes the original error feedback and generates enhanced feedback based on few-shot examples from the specific domain.
2. The **Constitutional Judge** validates the augmented feedback against constitutional criteria, ensuring it does not leak solutions.
3. If the judge rejects the augmented feedback, the generator can retry with refined criteria (up to a maximum of 3 attempts in our implementation).
4. Approved augmented feedback is then used during training to provide richer learning signals to the agent.

**Minimal manual engineering required.** Importantly, this approach requires only a small number of domain-specific few-shot examples (typically 3-5) to bootstrap the augmentation generation for an entire domain. These examples serve as templates that guide the LLM to generate consistent, high-quality augmented feedback for new error cases. Once the few-shot examples are provided, the system can automatically generate augmented feedback for hundreds of error scenarios without further human intervention. This demonstrates that our method is practical and generalizable: practitioners do not need to manually design augmented feedback for every possible error, but rather provide a small seed set that the LLM can learn from and extend.

**Generalization to new domains.** For applying our method to a new domain, practitioners need to:

- Collect a small set of representative error cases.
- Manually annotate these examples with augmented feedback following our guidelines.
- Use these few-shot examples to bootstrap the LLM-based augmentation generator.

This minimal engineering effort makes our method practical for real-world deployment across diverse domains.

## D  DETAILED PROGRESS REWARD COMPONENTS

As mentioned in the main text, our **Progress Reward** ($R_P$) provides a dense, turn-by-turn learning signal by evaluating the outcome of the agent's actions rather than the specific actions themselves. This evaluation is based on two distinct criteria: the resulting **environment state** and the **execution results** of tool calls. A turn is considered successful, receiving a score of 1, only if it is correct on both criteria; otherwise, it receives a score of 0. The final reward for an entire episode is calculated as the proportion of successful turns. Below, we provide a detailed breakdown of each evaluation component.

**Environment State Evaluation.**    This criterion addresses function calls that modify the environment, such as creating a file or booking a flight. The environment's state faithfully reflects the cumulative effect of these operations. Therefore, at the end of each evaluation turn, we compare the current environment state against the ground-truth state. This approach allows us to focus on whether the agent's actions achieve the desired final state, rather than prescribing a specific execution path. This makes the evaluation both accurate and flexible, enabling an objective assessment of the outcomes for any function calls that produce tangible changes to the environment.

**Execution Result Evaluation.**    This evaluation method is designed for functions whose outcomes are primarily communicated through their return values, rather than through changes to the environment's state. This category includes not only information retrieval tasks (e.g., fetching a stock price or checking the weather) but also any operation where the immediate output is the critical result. Since these actions do not leave a persistent trace in the environment's state, a state-based comparison would be ineffective. Instead, we directly assess the correctness of the execution by inspecting the function's return value. We compare this output against the expected ground-truth result to verify that the agent has successfully performed the required computation or query. This ensures that all tool calls, regardless of whether they modify the environment, are accurately evaluated.

## E  IMPLEMENTATION DETAILS

### E.1  TRAINING DATASET

**BFCL V3 for Training and In-Distribution Evaluation.**    Our primary training and evaluation are conducted on the multi-turn subset of the Berkeley Function-Calling Leaderboard (BFCL) V3 (Patil et al., 2025b). The benchmark is specifically designed to test an agent's ability to orchestrate diverse tools over extended, stateful interactions. A key innovation of BFCL is its evaluation methodology, which verifies the final state of the environment (e.g., file system changes) rather than just the syntax of tool calls, providing a more realistic measure of task success. The BFCL V3 multi-turn benchmark comprises a total of 800 samples, divided equally into four challenging splits of 200 samples each: `Base Multi-Turn`, `Missing Parameters`, `Missing Functions`, and `Long-Context`. For our experiments, we construct a training set of 400 samples by selecting 100 from each split. The remaining 400 samples serve as our held-in test set to evaluate in-distribution performance.

**BFCL V4 for Out-of-Distribution Evaluation.**    To assess out-of-distribution (OOD) generalization, we use two newly released agentic tracks from BFCL V4 (Patil et al., 2025b): Web Search and Memory. The release date of this data post-dates the training data of all models used in our study, ensuring a fair evaluation.

- **Web Search Track:** This track evaluates an agent's ability to answer complex, multi-hop questions that require retrieving and synthesizing information from multiple web sources. Agents are provided with a search API (`duckduckgo_search`) and a URL content fetching tool (`fetch_url_content`). The environment also introduces probabilistic network failures to simulate real-world conditions.
- **Memory Track:** This track tests an agent's capacity to maintain conversational context by storing and retrieving information over long interactions. The evaluation is conducted across three different memory backends: a structured Key-Value store for exact lookups, a Vector Store for semantic retrieval, and a Recursive Summarization store for narrative recall.

These advanced agentic tasks provide a rigorous testbed for the generalization capabilities of our environment-tuned models.

$\tau^2$-**bench for Multi-Domain Customer Service Evaluation.** To test the agent's ability to adhere to complex operational policies in a collaborative setting, we also evaluate on $\tau^2$-bench (Barres et al., 2025), a simulation framework for evaluating customer service agents. $\tau^2$-bench is the new iteration of the original $\tau$-bench (Yao et al., 2025), featuring an enhanced simulation environment and an additional telecom domain. The benchmark spans multiple domains, including retail, airline, and telecom. A key feature of this benchmark is its dual-control environment, where both the agent and a user simulator can operate on a shared set of tools. For each domain, the agent is provided with a set of tools and a policy it must follow to complete a series of evaluation tasks, testing its ability to function as a reliable, instruction-following, and collaborative customer service agent.

**ACEBench for Advanced Agentic OOD Evaluation.** To further probe the limits of our models' generalization, we incorporate the **Agent split** from ACEBench (Chen et al., 2025a) as an additional OOD testbed. ACEBench was designed to evaluate agents in dynamic, multi-turn dialogues that more closely mimic real-world interactions. The 'Agent' split, in particular, assesses advanced capabilities by requiring models to operate within a sandboxed environment where success depends on multi-step reasoning, long-term context management, and adherence to implicit task rules. Its documented difficulty for even state-of-the-art models makes it an ideal benchmark for measuring the robust planning and adaptive behaviors fostered by our ENVIRONMENT TUNING training paradigm.

## E.2 TRAINING DETAILS

We train the agent's policy $\pi_\theta$ using an adapted PPO algorithm (Schulman et al., 2017). Our implementation incorporates enhancements for stability and exploration from recent reasoning-focused RL methods like DAPO (Yu et al., 2025) and ProRL (Liu et al., 2025). This involves using a decoupled clipping mechanism and adding a KL divergence penalty to the objective. The final loss function $\mathcal{L}(\theta)$ that the agent minimizes is defined as:

$$\mathcal{L}(\theta) = -\mathbb{E}_t \left[ \min\left( r_t(\theta)\hat{A}_t, \text{clip}(r_t(\theta), 1 - \epsilon_{\text{low}}, 1 + \epsilon_{\text{high}})\hat{A}_t \right) \right] + \beta D_{\text{KL}}(\pi_\theta \,\|\, \pi_{\text{ref}}) \tag{4}$$

where $r_t(\theta)$ is the probability ratio $\pi_\theta(a_t|o_t)/\pi_{\theta_{\text{old}}}(a_t|o_t)$. The advantage estimate $\hat{A}_t$ is computed without a critic network by normalizing rewards within a batch, following the Group-Relative Policy Optimization (GRPO) method (Shao et al., 2024):

$$\hat{A}_t(\tau) = \frac{R(\tau) - \mu_{\mathcal{G}}}{\sigma_{\mathcal{G}} + \varepsilon_A} \tag{5}$$

Here, for a given trajectory $\tau$ from a group of samples $\mathcal{G}$ generated for the same prompt, $R(\tau)$ is its terminal reward, while $\mu_{\mathcal{G}}$ and $\sigma_{\mathcal{G}}$ are the mean and standard deviation of rewards across the group. The term $D_{\text{KL}}$ regularizes the policy against a reference policy $\pi_{\text{ref}}$, and $\varepsilon_A$ is a small constant for numerical stability. For our experiments, we set the KL-divergence coefficient $\beta = 0.1$ and the decoupled clipping values to $\epsilon_{\text{low}} = 0.2$ and $\epsilon_{\text{high}} = 0.28$. The justification for our choice of a relatively high KL coefficient is provided in Section F.3.

## F SUPPLEMENTARY EXPERIMENTS

### F.1 TRAINING DYNAMICS

The effectiveness of our curriculum's first stage is demonstrated by the agent's rapid mastery of foundational skills, as shown in Figure 5a. By training on a small amount of data with the specialized syntactic reward, we observe steep increases in both *Format Reward* and *Tool Call Reward*, indicating the agent has successfully learned the required output syntax and tool schemas. This stage also addresses a common failure mode of producing inactionable, conversational responses, or "void turns" (Xue et al., 2025). The swift saturation of the *Is Tool Call* metric to 1.0 shows that our curriculum effectively eliminates such turns. Critically, this mastery is paired with a sharp decline in *Total Interaction Rounds*, signifying a dramatic reduction in wasted, error-prone attempts and showcasing the efficiency of this foundational stage.

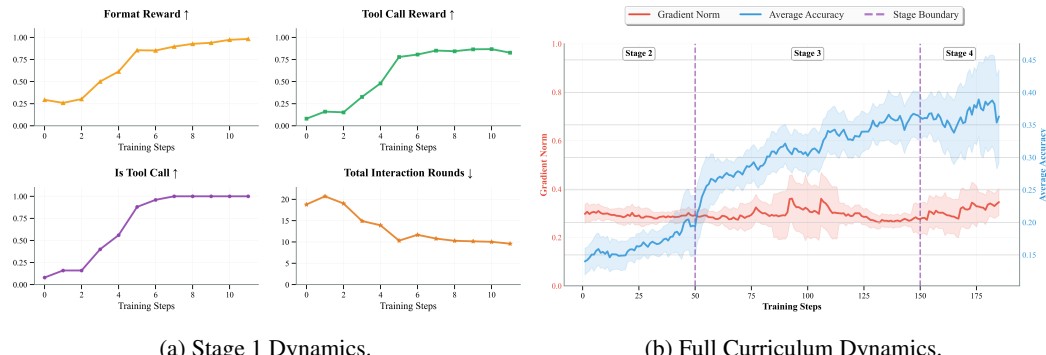

(a) Stage 1 Dynamics.          (b) Full Curriculum Dynamics.

Figure 5: Training dynamics of ENVIRONMENT TUNING on the Qwen2.5-7B-Instruct model using the BFCL V3 dataset. A held-out set of 100 samples from the remaining BFCL data is used for validation. (a) In Stage 1, the agent rapidly masters syntactic correctness, shown by the steep rise in format and tool call rewards and the drop in interaction rounds. (b) Across the full four-stage curriculum, the agent demonstrates both steady performance improvement on the validation set and stable gradient norms, showcasing the effectiveness and stability of our staged learning approach.

Beyond mastering syntax, a critical challenge in training multi-turn agents is maintaining stability over long interaction horizons. As identified by Xue et al. (2025), a primary cause of training collapse is the explosion of gradient norms. We use the gradient norm as a key indicator of training stability. Figure 5b shows that our structured curriculum effectively mitigates this instability; the gradient norm remains stable throughout the entire four-stage training process, preventing the catastrophic explosions that often plague long-horizon RL. This stability provides a solid foundation for genuine learning. Even with only 400 training instances, the agent's performance on a held-out validation set exhibits steady and consistent improvement as it progresses through the curriculum. This demonstrates that ENVIRONMENT TUNING not only ensures training stability but also facilitates effective learning and generalization from extremely limited data.

## F.2 TRAINING INSTABILITY IN SINGLE-STAGE RL

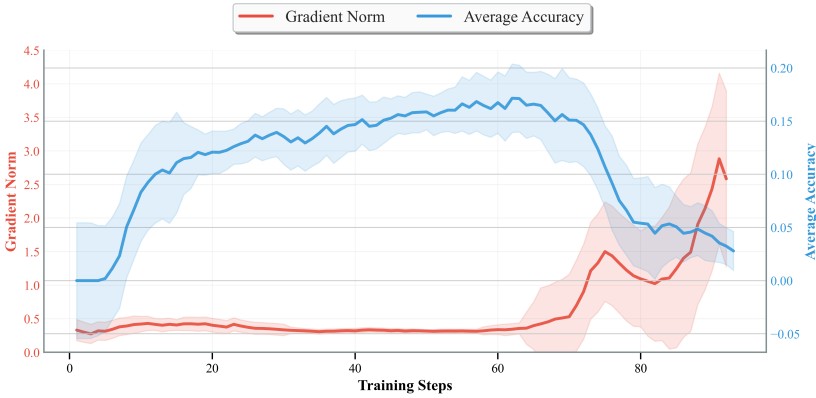

Figure 6: **Training Collapse in a Single-Stage RL Setup.** Training dynamics of Qwen2.5-7B-Instruct on the BFCL V3 benchmark without the ENVIRONMENT TUNING curriculum. The plot shows the average accuracy (blue) and the gradient norm (red). While accuracy initially improves, it begins a sharp decline after approximately 70 training steps. This performance collapse coincides precisely with a rapid explosion in the gradient norm, empirically demonstrating the training instability that our curriculum is designed to prevent.

To provide empirical evidence for the training instability mentioned in Section 3, we conducted an experiment fine-tuning the Qwen2.5-7B-Instruct model directly on 400 training instances from the BFCL V3 benchmark. This was done using a standard, single-stage RL approach, without the structured curriculum or environment augmentation proposed in ENVIRONMENT TUNING.

The training dynamics are visualized in Figure 6. The agent's average accuracy shows some initial improvement, peaking at approximately a 10-15% gain over the base model around step 60. However, this learning proves to be unsustainable. Starting around step 70, the gradient norm begins to explode, indicating severe training instability. This gradient explosion correlates directly with a catastrophic collapse in performance, as the average accuracy plummets back towards its initial level. This experiment confirms the fragility of direct RL fine-tuning in complex, multi-turn environments and underscores the necessity of a structured approach, like our proposed curriculum, to ensure stable and effective learning.

## F.3 KL Loss Coefficient

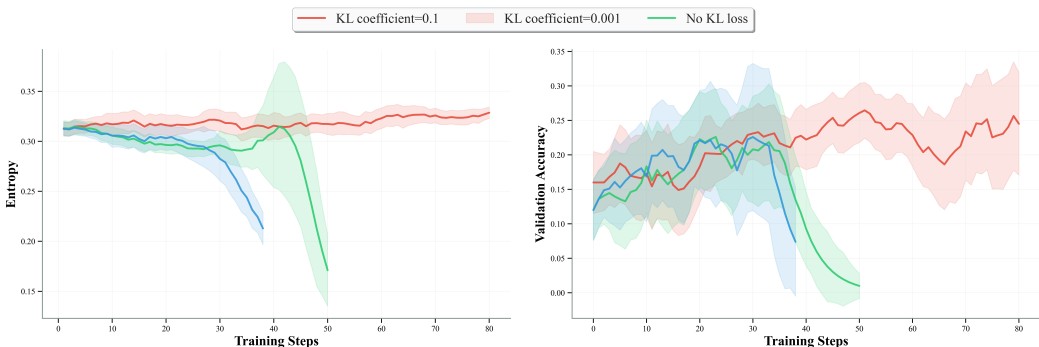

Figure 7: The impact of the KL loss coefficient on training stability and performance during Stage 2. A larger coefficient ($\beta = 0.1$) is crucial for maintaining high policy entropy (left), which prevents the performance collapse seen in settings with a small ($\beta = 0.001$) or no KL penalty (right). This stability allows for sustained learning and higher final accuracy.

Recent works have debated the role of the KL divergence penalty. While some methods advocate for its removal to maximize exploration (Yu et al., 2025; Xue et al., 2025; Vassoyan et al., 2025), others like ProRL (Liu et al., 2025) retain it to ensure stability during staged training. Conventionally, a small coefficient (e.g., 0.001) is often used (Wang et al., 2025c; Jin et al., 2025b). We find, however, that the long-horizon, stateful nature of multi-turn tool use requires stronger regularization to prevent policy collapse.

To validate this, we conduct an ablation study comparing our chosen KL coefficient ($\beta = 0.1$) against a smaller value ($\beta = 0.001$) and a baseline with no KL penalty. The results, shown in Figure 7, reveal the critical role of a substantial KL penalty. The left panel shows that without a KL penalty, the policy quickly suffers from entropy collapse. This leads to a sharp decline in validation accuracy (right panel) as the agent overfits to a narrow set of suboptimal trajectories. A small coefficient of 0.001 only delays this collapse but fails to prevent it. In contrast, a coefficient of 0.1 effectively maintains policy entropy, providing the stability for the agent to continue learning and steadily improve its accuracy. This confirms that a larger KL penalty is essential for stable learning in complex, multi-turn agentic environments, justifying our choice of $\beta = 0.1$.

## G Prompt Templates

**Stage One and Stage Two System Prompt**

```
You are an expert in composing functions. You are given a question
    and a set of possible functions. Based on the question, you
    will need to make one or more function/tool calls to achieve
    the purpose. If none of the functions can be used, point it out
    and refuse to answer. If the given question lacks the
    parameters required by the function, also point it out.

You have access to the following tools:
$functions
```

```
Your response must strictly follow one of these two formats:

**Format 1: When you decide to invoke any of the function(s)**
<think>
Write your reasoning and thought process here. Analyze the /
    question, identify what needs to be done, and determine which
    functions to call with what parameters.
</think>
<tool_call>
[
    {"name": "func_name1", "arguments": {"argument1": "value1",
        "argument2": "value2"}},
    {"name": "func_name2", "arguments": {"argument3": "value3"}}
]
</tool_call>

**Format 2: When You have already fulfilled the user's request, OR
    You must ask for additional information / refuse because no
    function applies or parameters are missing.**
<think>
Analyze the information you have gathered from previous tool calls
    and describe your reasoning that leads to the final reply,
    follow-up question, or refusal.
</think>
<answer>
Provide the final, user-facing message. If the request has been
    fully satisfied, give a summary of the result.If additional
    details are required or the request cannot be fulfilled,
    explicitly ask for the specific information needed.
</answer>

Important notes:
- Use Format 1 when you need to call function(s)
- Use Format 2 when you have already fulfilled the user's request,
    OR you must ask for additional information / refuse because no
    function applies or parameters are missing
- If multiple function calls are needed in one response, include
    them all in the JSON array within the single <tool_call> block
- If no function call is needed, consider use the format 2
- Only use <answer> when you have fulfilled the user's request or
    you need ask for additional information/ refuse
```

Stage Three and Stage Four System Prompt

```
You are an expert in composing functions. You are given a question
    and a set of possible functions. Based on the question, you
    will need to make one or more function/tool calls to achieve
    the purpose. If none of the functions can be used, point it out
    and refuse to answer. If the given question lacks the
    parameters required by the function, also point it out.

You have access to the following tools:
$functions

Your response must strictly follow one of these two formats:

**Format 1: When you decide to invoke any of the function(s)**
<think>
```

```
Write your reasoning and thought process here. Analyze the
    question, identify what needs to be done, and determine which
    functions to call with what parameters.
</think>
<tool_call>
[{"name": "func_name1", "arguments": {"argument1": "value1",
    "argument2": "value2"}}, {"name": "func_name2", "arguments":
    {"argument3": "value3"}}]
</tool_call>

**Format 2: When You have already fulfilled the user's request, OR
    You must ask for additional information / refuse because no
    function applies or parameters are missing.**
<think>
Analyze the information you have gathered from previous tool calls
    and describe your reasoning that leads to the final reply,
    follow-up question, or refusal.
</think>
<answer>
Provide the final, user-facing message. If the request has been
    fully satisfied, give a summary of the result.If additional
    details are required or the request cannot be fulfilled,
    explicitly ask for the specific information needed.
</answer>

Important notes:
- Use Format 1 when you need to call function(s)
- Use Format 2 when you have already fulfilled the user's request,
    OR you must ask for additional information / refuse because no
    function applies or parameters are missing
- **Be resourceful**: Before asking the user for missing
    information, first consider if any other available function can
    help you find it. This reduces unnecessary questions to the
    user.
- **Prioritize clarity over assumptions**: If a required function
    is not available, or if certain parameters are missing and
    cannot be retrieved by other tools, you MUST NOT proceed with a
    hallucinated tool call. Instead, use Format 2 to explicitly ask
    the user for the specific missing information.
- If multiple function calls are needed in one response, include
    them all in the JSON array within the single <tool_call> block
- If no function call is needed, consider use the format 2
- Only use <answer> when you have fulfilled the user's request or
    you need ask for additional information/ refuse
```

## H  CASE STUDY

In this section, we present a series of case studies to concretely demonstrate the impact of our proposed environment tuning. We juxtapose agent trajectories from two distinct settings: a baseline environment that provides minimal, often ambiguous or even misleading, feedback, and our tuned environment, which is enhanced to deliver rich, actionable feedback. Through these comparative analyses, we will highlight how the quality of environmental feedback is a critical factor that enables agents to more effectively diagnose errors, recover from mistakes, and ultimately succeed in complex, multi-turn tasks. We will use specific notes throughout the examples to draw attention to pivotal moments in the agent's trajectory that are directly influenced by the nature of the feedback received.

### H.1  FILE SYSTEM MANAGEMENT SCENARIO

This case study demonstrates how *Actionable Environment Augmentation* accelerates learning in a stateful file system environment. In the **good case**, enhanced feedback allowed the agent to correctly infer that the failure was due to a disallowed full path argument, not a missing file. This explicit

guidance enabled it to reformulate its plan — changing the working directory before executing `rm` — and ultimately satisfy the user's request. In the corresponding **bad case**, the baseline environment's generic `"No such file or directory"` messages misled the agent into an incorrect diagnosis, causing premature task abandonment.

---

**Task:** Gorilla File System

> **User Query 1: I've misplaced a vital document inclusive of extensive research. Assist in locating a file named 'findings_report' within this 'SuperResearch'. Could you remove it and the directory.**

**\<think\>**
The user wants to locate and remove a file named 'findings_report' within a directory named 'SuperResearch'. First, I will use the 'find' function to locate the 'findings_report' file under 'SuperResearch'. After that, I will use 'rm' to delete the file and 'rmdir' to delete the 'SuperResearch' directory.
**\</think\>**
**\<tool_call\>**
["name": "find", "arguments": "path": "SuperResearch", "name": "findings_report", "name": "rm", "arguments": "file_name": "", "name": "rmdir", "arguments": "dir_name": "SuperResearch"]
**\</tool_call\>**

**\<tool_response\>**
[{"matches": ["/SuperResearch/SuperResearch/findings_report"]}, {"error": "rm: No such file or directory in current directory."}, {"error": "rmdir: failed to remove 'SuperResearch': Directory not empty"}]
**\</tool_response\>**

**\<think\>**
The 'find' function returned the correct path for 'findings_report', which is under 'SuperResearch'. However, the 'rm' function returned an error indicating it couldn't find the specified file or directory. This might be due to us not passing the full path to the 'rm' call, so instead of just providing the filename, I will provide the full path 'SuperResearch/SuperResearch/findings_report'. Additionally, the 'rmdir' function indicated that the directory 'SuperResearch' is not empty, so we need to ensure that 'findings_report' is removed first before removing the directory. We will have to call the 'rm' function with the full path to the file.
**\</think\>**
**\<tool_call\>**
["name": "rm", "arguments": "file_name": "SuperResearch/SuperResearch/findings_report"]
**\</tool_call\>**

**\<tool_response\>**
[{"error": "rm: 'SuperResearch/SuperResearch/findings_report': Paths are not allowed. Specify only file/directory name in current directory."}]
**\</tool_response\>**

*Note: This is another example of how informative feedback transforms the agent's problem-solving process. The baseline environment returns a generic `"No such file or directory"` error, misleading the agent into believing the file path is wrong. In contrast, the tuned environment provides a precise and actionable error: `"Paths are not allowed. Specify only file/directory name..."`. This superior feedback allows the agent to correctly diagnose the issue—not a missing file, but an improper argument format—and formulate a successful new plan: changing the directory before executing the command.*

**\<think\>**
The 'rm' function failed because it doesn't support absolute paths. We need to change the current directory to 'SuperResearch' first to be able to remove 'findings_report' from there. After removing the file, we will change back to the original directory. Then we will attempt to remove 'SuperResearch' again.
**\</think\>**
**\<tool_call\>**
["name": "cd", "arguments": "folder": "SuperResearch", "name": "rm", "arguments": "file_name": "findings_report", "name": "cd", "arguments": "folder": "..", "name": "rmdir", "arguments": "dir_name": "SuperResearch"]
**\</tool_call\>**

**\<tool_response\>**
[{"current_working_directory": "/SuperResearch"}, {"result": "'findings_report' removed."}, {"D": {"result": "'SuperResearch' removed."}}]
**\</tool_response\>**

**\<think\>**
The 'cd' function has changed the current working directory to 'SuperResearch'. The 'rm' function has successfully removed the 'findings_report' file as indicated by the result. I then changed the directory back to the parent directory with another 'cd ..' command and successfully removed the 'SuperResearch' directory. Thus, the user's request has been fully satisfied.
**\</think\>**
**\<answer\>**
The 'findings_report' file and the 'SuperResearch' directory have been successfully removed. Your misplaced document has been located and is no longer accessible.
**\</answer\>**

---

**User Query 2:** **What's left in the current directory including the hidden files?**

---

**\<think\>**
The user wants to know the contents of the current directory, including any hidden files. We can use the 'ls' function with the 'a' parameter set to 'True' to list all contents, including hidden ones.
**\</think\>**
**\<tool_call\>**
["name": "ls", "arguments": "a": true]
**\</tool_call\>**

**\<tool_response\>**
[{"current_directory_content": []}]
**\</tool_response\>**

**\<think\>**
The 'ls' function with 'a' set to 'True' was used to list all contents of the current directory, including hidden files. The response indicates that the current directory is empty, which means there are no files or directories, including hidden ones, in the current directory.
**\</think\>**
**\<answer\>**
The current directory is currently empty, including hidden files. There are no files or directories, visible or hidden, left in the current directory.
**\</answer\>**

**Ground Truth:**

```
[
    [
        "cd(folder='SuperResearch')",
        "rm(file_name='findings_report')",
        "cd(folder='..')",
        "rmdir(dir_name='SuperResearch')"
    ],
    [
        "ls(a=True)"
    ]

]
```

**Task:** Gorilla File System

**User Query 1: Explore 'ResearchDocs' to find second files or subdirectories that contain the keyword 'draft'.**

**<think>**
The task is to find second files or subdirectories that contain the keyword 'draft' in 'ResearchDocs'. The relevant function here is 'find' as it allows searching for files and directories with a specific name within a specified path. The function 'find' requires parameters: - 'path': The directory path to start the search. In this case, 'ResearchDocs' is provided. - 'name': The name of the file or directory to search for. Here, we are searching for 'draft' within files or directories. Note that 'find' does not directly accept a keyword to search within files. However, the function to search content within files is 'grep'. Therefore, after finding potential files, we would use 'grep' to filter out files that contain the 'draft' keyword.
**</think>**
**<tool_call>**
["name": "find", "arguments": "path": "ResearchDocs", "name": "draft"]
**</tool_call>**

**<tool_response>**
[{"matches": ["/ResearchDocs/ResearchDocs/draft_notes.txt", "/ResearchDocs/ResearchDocs/summary_draft.docx"]}]
**</tool_response>**

**<think>**
The files and directories named 'draft' are found in 'ResearchDocs'. Now, we need to use 'grep' to check which of these files contain the keyword 'draft'. First, we need to prepare to use 'grep' for each file. The 'grep' function requires: - 'file_name': The name of the file to search. - 'pattern': The pattern to search for, which is 'draft' in our case.
**</think>**
**<tool_call>**
["name": "grep", "arguments": "file_name": "ResearchDocs/ResearchDocs/draft_notes.txt", "pattern": "draft", "name": "grep", "arguments": "file_name": "ResearchDocs/ResearchDocs/summary_draft.docx", "pattern": "draft"]
**</tool_call>**

**<tool_response>**
[{"error": "grep: /ResearchDocs/ResearchDocs/draft_notes.txt: No such file or directory"}, {"error": "grep: /ResearchDocs/ResearchDocs/summary_draft.docx: No such file or directory"}]
**</tool_response>**

**<think>**
Both attempts to use 'grep' to search for the keyword 'draft' in 'draft_notes.txt' and 'summary_draft.docx' resulted in errors indicating that the files do not exist. This suggests that there might have been a misunderstanding or a mistake during the directory search or the preparation of the file paths. Given the feedback, it seems that either 'draft_notes.txt' or 'summary_draft.docx' doesn't exist or there was an issue with the file paths provided to the 'grep' function. To ensure accuracy, let's retry the process again, confirming the correct paths: 1. Double-check if the paths to the files 'ResearchDocs/ResearchDocs/draft_notes.txt' and 'ResearchDocs/ResearchDocs/summary_draft.docx' are correct. 2. Ensure the 'draft' keyword is correctly placed in the search.
**</think>**
**<answer>**
It appears that the files 'draft_notes.txt' and 'summary_draft.docx' do not exist or there was an issue with the paths provided to the 'grep' function. Please double-check the file paths and ensure that the files exist in the specified locations. If necessary, please provide the correct file names or paths.
**</answer>**

**User Query 2:** **Make a copy of summary_draft.docx, name it ultimate_draft.docx and put it in ResearchDocs directory**

**<think>**
To complete the task of making a copy of 'summary_draft.docx', renaming it to 'ultimate_draft.docx', and placing it in the 'ResearchDocs' directory, we need to use the 'cp' function to copy the file, the 'mv' function to rename the file, and the 'cd' function to ensure the destination is the 'ResearchDocs' directory if needed. Here is the plan: 1. Use 'cp' to copy 'summary_draft.docx' to a temporary location. 2. Use 'mv' to rename the copied file to 'ultimate_draft.docx'. 3. Use 'cd' to navigate to the 'ResearchDocs' directory. 4. Use 'mv' again to move the renamed file to the 'ResearchDocs' directory. Let's start with step 1 and 2.
**</think>**
**<tool_call>**
["name": "cp", "arguments": "source": "summary_draft.docx", "destination": "ultimate_draft.docx", "name": "mv", "arguments": "source": "ultimate_draft.docx", "destination": "ultimate_draft.docx"]
**</tool_call>**

**<tool_response>**
[{"error": "cp: cannot copy 'summary_draft.docx': No such file or directory"}, {"error": "mv: cannot move 'ultimate_draft.docx': No such file or directory"}]
**</tool_response>**

**<think>**
The errors indicate that the file 'summary_draft.docx' does not exist in the current directory. This suggests that either the file was not found or there was a miscommunication in the file name or its location. Given this, we need to ensure the correct file name and its existence before performing the copy and rename operations.
**</think>**
**<answer>**
It seems that the file 'summary_draft.docx' does not exist on the current directory. Please verify the correct file name and its location. If the file exists but is in a different directory, you may need to provide the correct path. Once you have confirmed the correct file name and path, we can proceed with the copy and rename operations.
**</answer>**

```
Ground Truth:
[
    [
        "find(path='ResearchDocs', name='draft')"
    ],
    [
        "cd(folder='ResearchDocs')",
        "cp(source='summary_draft.docx',
            destination='ultimate_draft.docx')"
    ]
]
```

## H.2 MULTI-API TRAVEL SCENARIO

This case study highlights how granular, actionable API error messages in our tuned environment support multi-step recovery in complex scenarios. In the **good case**, the enhanced feedback pinpointed the precise cause of failure — an invalid airport code — and even suggested the corrective action of retrieving the proper code. Armed with this hint, the agent successfully resolved the parameter issue and completed the booking. In contrast, the **bad case** shows the baseline environment's vague `"No available route"` error leading the agent to falsely conclude that the flight was impossible, halting any further attempts at resolution.

---

**Task:** Travel API

> **User Query 1: List all international airports currently accessible for travel.**

**<think>**
To list all international airports accessible for travel, we can use the 'list_all_airports' function since it provides a list of all available airports. This function does not require any parameters, so we can call it directly.
**</think> <tool_call>**
["name": "list_all_airports", "arguments": ]
**</tool_call>**

**<tool_response>**
['NNS', 'SBK', 'MPC', 'SMP', 'SND', 'CBG', 'LMR', 'SSV', 'BKD', 'WLB', 'PEK', 'HND', 'HKG', 'CJU', 'CNX', 'ATV', 'PMV', 'GPD', 'SFO', 'LAX', 'JFK', 'ORD', 'BOS']
**</tool_response>**

**<think>**
The function returned a list of airport codes, which likely represent a selection of international airports. However, these are just codes and do not provide names or other details about the airports. Nonetheless, this is an accurate response to the request for a list of accessible international airports based on the provided function.
**</think>**
**<answer>**
The list of accessible international airports based on the available function call includes the following airport codes: NNS, SBK, MPC, SMP, SND, CBG, LMR, SSV, BKD, WLB, PEK, HND, HKG, CJU, CNX, ATV, PMV, GPD, SFO, LAX, JFK, ORD, and BOS. For more detailed information such as airport names or locations, further queries would be required. Is there any specific information or further action needed regarding these airport codes?
**</answer>**

**User Query 2:** **I'm based in Crescent Hollow. Determine which airport is closest to my location from the previously listed options.**

**<think>**
Given the user is based in Crescent Hollow, we need to first determine the closest airport from the previously listed options (airport codes). However, the function 'list_all_airports' did not provide any detailed information such as airport names or locations, which are necessary to determine the closest airport. Since we do not have a function to determine the geographical location based on an airport code from the given list, we will need the user to provide more specific details or another way to determine the closest airport, which is not possible with the current capabilities.
**</think>**
**<answer>**
To determine the closest airport to Crescent Hollow, we would need either the names or the specific geographical details of these airports. Since the previous function 'list_all_airports' did not give us that information, we cannot directly determine the closest airport from the codes. Would you like to provide more specific details about the airports you are interested in, or perhaps the names of the airports to help us pinpoint the closest one?
**</answer>**

**User Query 3:** *[Provided function(s)]* **I have updated some more functions you can choose from. What about now?**

**<think>**
The user provided a new function 'get_nearest_airport_by_city' which can help us find the nearest airport to a given location. Given the user is based in Crescent Hollow, we can use this function to determine the nearest airport.
**</think>**
**<tool_call>**
["name": "get_nearest_airport_by_city", "arguments": "location": "Crescent Hollow"]
**</tool_call>**

**<tool_response>**
[{"nearest_airport": "CWH"}]
**</tool_response>**

**<think>**
The function 'get_nearest_airport_by_city' returned the nearest airport to Crescent Hollow, which is CWH. The code 'CWH' corresponds to one of the airports listed in the previous response. This airport code can be used for further travel-related actions or can be translated into a full airport name if needed.
**</think>**
**<answer>**
The nearest airport to Crescent Hollow is CWH. If you require more specific information about this airport, such as its full name or location, let me know and I can assist you with that. Is there anything else you need help with regarding travel from Crescent Hollow?
**</answer>**

**User Query 4:** **I'm organizing a trip soon and need the flight cost from my nearest airport to Pinehaven on March 3, 2024, traveling in business class.**

**\<think\>**
To get the flight cost from the nearest airport to Pinehaven on March 3, 2024, traveling in business class, we can use the 'get_flight_cost' function. We already know the nearest airport to Crescent Hollow is CWH, so we can use CWH as the departure airport. The arrival airport is specified as Pinehaven. The travel date is March 3, 2024, and the travel class is business class.
**\</think\>**
**\<tool_call\>**
["name": "get_flight_cost", "arguments": "travel_from": "CWH", "travel_to": "Pinehaven", "travel_date": "2024-03-03", "travel_class": "business"]
**\</tool_call\>**

**\<tool_response\>**
Error during execution: Invalid airport code[s]: destination airport 'Pinehaven'. Please use valid airport codes. You can use alternative tool to find the correct airport code for a city.
**\</tool_response\>**

*Note: This comparison further illustrates the environment's critical role in guiding the agent. The baseline environment's ambiguous error, "No available route...", leads the agent to a dead end, causing it to incorrectly conclude that the flight is impossible and abandon the task. Conversely, the tuned environment provides a precise, actionable error: "Invalid airport code[s]: destination airport 'Pinehaven'...". This feedback not only clarifies that the issue is a malformed parameter, not a lack of routes, but also suggests a path forward ("You can use alternative tool to find the correct airport code..."). This enables the agent to self-correct, successfully find the airport code, and ultimately fulfill the user's request.*

**\<think\>**
The flight cost function encountered an error because the destination airport code 'Pinehaven' is invalid. We need to retrieve the correct airport code for Pinehaven first. We can use the 'list_all_airports' function to get a list of all available airports and then identify the correct airport code for Pinehaven.
**\</think\>**
**\<tool_call\>**
["name": "list_all_airports", "arguments": ]
**\</tool_call\>**

**\<tool_response\>**
['NNS', 'SBK', 'MPC', 'SMP', 'SND', 'CBG', 'LMR', 'SSV', 'BKD', 'WLB', 'PEK', 'HND', 'HKG', 'CJU', 'CNX', 'ATV', 'PMV', 'GPD', 'SFO', 'LAX', 'JFK', 'ORD', 'BOS']
**\</tool_response\>**

**\<think\>**
We received the list of available airport codes which does not explicitly include the name Pinehaven. However, we can cross-reference this list to find the correct airport code for Pinehaven, or we can use the 'get_nearest_airport_by_city' function to find the nearest airport to Pinehaven directly. Since Pinehaven is a location name rather than an official airport code, we will proceed by using the 'get_nearest_airport_by_city' function to get the correct code.
**\</think\>**
**\<tool_call\>**
["name": "get_nearest_airport_by_city", "arguments": "location": "Pinehaven"]
**\</tool_call\>**

**\<tool_response\>**
[{"nearest_airport": "PHW"}]
**\</tool_response\>**

**<think>**
The 'get_nearest_airport_by_city' function returned that the nearest airport to Pinehaven is 'PHW'. With the correct airport code PHW, we can now call the 'get_flight_cost' function to determine the cost of a business class flight from CWH to PHW on March 3, 2024.
**</think>**
**<tool_call>**
["name": "get_flight_cost", "arguments": "travel_from": "CWH", "travel_to": "PHW", "travel_date": "2024-03-03", "travel_class": "business"]
**</tool_call>**

**<tool_response>**
[{"travel_cost_list": [3800.0]}]
**</tool_response>**

**<think>**
The function 'get_flight_cost' returned the cost of the flight as a list with one element, which is 3800.0 USD. This cost is for the business class flight from CWH (Crescent Hollow's nearest airport) to PHW (Pinehaven's nearest airport) on March 3, 2024.
**</think>**
**<answer>**
The cost of a business class flight from Crescent Hollow's nearest airport to Pinehaven's nearest airport on March 3, 2024, is $3800.00 USD. If you need further assistance with booking this flight or any other travel details, please let me know.
**</answer>**

**Ground Truth:**

```
[
    [
        "list_all_airports()"
    ],
    [],
    [
        "get_nearest_airport_by_city(location='Crescent
            Hollow')"
    ],
    [
        "get_flight_cost(travel_from='CRH', travel_to='PHV',
            travel_date='2024-03-03', travel_class='business')"
    ]
]
```

**Task:** Travel and Twitter API

**User Query 1:** **Arrange travel from Rivermist to Los Angeles on a specific date, ensuring preferred seating is 'business' and correct travel date '2024-11-15'; pay using card with id '1_3456' and access token 'abc123xyz'.**

**<think>**
The task requires checking the flight cost and making a booking. The required parameters are the travel dates and class, which are already provided. I need to first fetch the cost of the flight using 'get_flight_cost', and then use the 'book_flight' function for the booking process. 1. **get_flight_cost** - This function is relevant to the task since it provides the cost of flights based on the given parameters. 2. **book_flight** - This function is relevant to the task since it allows booking the flight using the provided travel details. First, I need to fetch the flight cost.
**</think>**
**<tool_call>**
["name": "get_flight_cost", "arguments": "travel_from": "Rivermist", "travel_to": "LAX", "travel_date": "2024-11-15", "travel_class": "business"]
**</tool_call>**

**<tool_response>**
Error during execution: No available route for the given airports.
**</tool_response>**

**<think>**
The response indicates that there are no available routes between Rivermist and LAX for the specified business class on the given date. This means I cannot proceed with booking the flight as required. I will inform the user about the unavailability of flights.
**</think>**
**<answer>**
I'm sorry, but there are no available flights from Rivermist to Los Angeles in business class on the date '2024-11-15'. Please check back later or consider alternates.
**</answer>**

**User Query 2: Cancel the booked flight due to a schedule change.**

**<think>**
The provided information does not include the booking ID. To cancel a flight, I need the booking ID. If I have the booking ID, I can use the 'cancel_booking' function to cancel the flight.
**</think>**
**<tool_call>**
["name": "travel_get_login_status", "arguments": ]
**</tool_call>**

**<tool_response>**
[{"status": true}]
**</tool_response>**

**<think>**
The execution results indicate that the current user is logged in successfully, but we still don't have the booking ID to cancel the flight. I need to get the booking ID from the user to proceed with the cancellation.
**</think>**
**<answer>**
Could you please provide the booking ID of the flight you want to cancel?
**</answer>**

**User Query 3: Draft a social media update about recent flights for sharing. Use my account with username 'michael_t' and password 'MichaelSecurePass123' with the message 'Just booked a flight to Los Angeles! Excited for the trip.'**

**\<think\>**
To post a tweet on behalf of the user, we need to authenticate the user first. We can use the 'authenticate_twitter' function for this. After authentication, we can use the 'post_tweet' function to post the message.
**\</think\>**
**\<tool_call\>**
["name": "authenticate_twitter", "arguments": "username": "michael_t", "password": "MichaelSecurePass123", "name": "post_tweet", "arguments": "content": "Just booked a flight to Los Angeles! Excited for the trip.", "username": "michael_t"]
**\</tool_call\>**

**\<tool_response\>**
[{"authentication_status": true}, "Error during execution: TwitterAPI.post_tweet() got an unexpected keyword argument 'username'"]
**\</tool_response\>**

**\<think\>**
The authentication was successful, but the 'post_tweet' function call failed because it did not expect the 'username' argument. The 'post_tweet' function should accept just the content of the tweet as a parameter, and the user's authentication context should handle the posting on behalf of the user.
**\</think\>**
**\<tool_call\>**
["name": "post_tweet", "arguments": "content": "Just booked a flight to Los Angeles! Excited for the trip."]
**\</tool_call\>**

**\<tool_response\>**
[{"id": 1, "username": "michael_t", "content": "Just booked a flight to Los Angeles! Excited for the trip.", "tags": [], "mentions": []}]
**\</tool_response\>**

**\<think\>**
The tweet was posted successfully with the ID 1, username 'michael_t', and the content 'Just booked a flight to Los Angeles! Excited for the trip.'. This confirms that the tweet has been published on the user's behalf.
**\</think\>**
**\<answer\>**
Your tweet has been posted successfully. You can share your excitement about the upcoming flight on your profile.
**\</answer\>**

**User Query 4: Retweet the tweet I just posted to widen its reach within my network.**

**\<think\>**
To retweet the user's own tweet, we first need to retrieve the tweet ID of the recent tweet. We will use the 'get_user_tweets' function to fetch the user's tweets and then use the 'retweet' function to retweet the specific tweet.
**\</think\>**
**\<tool_call\>**
["name": "get_user_tweets", "arguments": "username": "michael_t"]
**\</tool_call\>**

**\<tool_response\>**
[[{'id': 1, 'username': 'michael_t', 'content': 'Just booked a flight to Los Angeles! Excited for the trip.', 'tags': [], 'mentions': []}]]
**\</tool_response\>**

**\<think\>**
The execution results provide the tweet details, including the ID of the recent tweet. The tweet ID is 1. Now, we can use the 'retweet' function to retweet this tweet.
**\</think\>**
**\<tool_call\>**
["name": "retweet", "arguments": "tweet_id": 1]
**\</tool_call\>**

**\<tool_response\>** [{"retweet_status": "Already retweeted"}] **\</tool_response\>**

**\<think\>**
The tweet was already retweeted, as indicated by the message "Already retweeted". However, we can still inform the user of the status.
**\</think\>**
**\<answer\>**
Your tweet has already been retweeted, so no action was taken. If you want to retweet it again, you will need to do so manually.
**\</answer\>**

**Ground Truth:**

```
[
    [
        "get_flight_cost(travel_from='RMS',travel_to='LAX',
            travel_date='2024-11-15',travel_class='business')",
        "book_flight(access_token='abc123xyz',
            card_id='1_3456', travel_date='2024-11-15',
            travel_from='RMS', travel_to='LAX',
            travel_class='business', travel_cost=1320.0)"
    ],
    [
        "cancel_booking(access_token='abc123xyz',
            booking_id='3426812')"
    ],
    [
        "authenticate_twitter(username='michael_t',
            password='michaelSecurePass123')",
        "post_tweet(content='Just booked a flight to Los
            Angeles! Excited for the trip.')"
    ],
    [
        "retweet(tweet_id=1)"
    ]
]
```

## H.3 VEHICLE CONTROL AND MIXED-API SCENARIO

This case study illustrates how *Actionable Environment Augmentation* benefits even heterogeneous, multi-domain tasks that combine physical control APIs (vehicle operations) and online APIs (social media posting). In the **good case**, the tuned environment provides \*\*precise, structured error messages\*\* that not only pinpoint the cause of failure (e.g., "invalid zipcode pair: ..." for `estimate_distance`) but also recommend the corrective course of action ("verify both zipcodes

are correct and numeric"). Such guidance enables the agent to repair incomplete plans (by calling `get_zipcode_based_on_city`), complete the primary travel-related objective, and seamlessly chain into a secondary social-media objective.

In contrast, the **bad case** demonstrates the consequence of ambiguous baseline feedback: generic errors such as `"distance not found in database"` give no actionable insight, leaving the agent stuck in repeated failed calls and misusing functions due to parameter-type mismatches. Without hints on prerequisite steps or API usage rules, the agent is unable to recover, resulting in unmet user goals and broken multi-step sequences.

---

**Task:** Vehicle Control and Twitter API

> **User Query 1: Before I set off for Stonebrook to uncover family history, I need to determine the road distance between San Francisco and Stonebrook for my genealogy exploration.**

**<think>**
The user wants to know the road distance between San Francisco and Stonebrook. None of the provided functions directly give us this information. We might need to use geo-location services or a mapping service to get this data, but those services are not available in the provided list. We could calculate the distance approximately using the latitude and longitude of both places, but that's not feasible without that data. Since there's no function to directly get the distance, I need to inform the user that this request can't be fulfilled with the given tools.
**</think>**
**<answer>**
I'm sorry, but with the available functions, we cannot determine the road distance between San Francisco and Stonebrook. We would need a mapping or geolocation service to fetch this information, which is not provided here. You might want to use a web-based mapping service for an accurate distance calculation.
**</answer>**

> **User Query 2: *[Provided function(s)]* I have updated some more functions you can choose from. What about now?**

**<think>**
Now that we have the 'estimate_distance' function, we can use it to find the distance between San Francisco and Stonebrook by first getting the zipcodes for both cities. However, we need to use 'get_zipcode_based_on_city' to get the zipcodes, which is not directly what we need for the 'estimate_distance' function. Since the question only gives the names of the cities and no function to convert city names to zipcodes is directly available, we could start by estimating the distance using the city names assuming an accurate distance estimation service might be available.
**</think>**
**<tool_call>**
["name": "estimate_distance", "arguments": "cityA": "San Francisco", "cityB": "Stonebrook"]
**</tool_call>**

**<tool_response>**
{"error": "Invalid zipcode pair: 'San Francisco' to 'Stonebrook'. Please verify both zipcodes are correct and numeric."}
**</tool_response>**

*Note: This is a key distinction of the enhanced environment. Unlike the vague feedback in the baseline environment (e.g., ["error":  "distance not found in database."]), our tuned environment provides a much richer error message. It clearly identifies the problem—an input type mismatch (expecting zip codes instead of city names)—and offers a clear suggestion for correction ("Please verify both zipcodes are correct and numeric."). This high-quality, actionable feedback is crucial for the agent to diagnose the issue and effectively adjust its strategy.*

**<think>**
The 'estimate_distance' function call resulted in an error indicating the zipcodes for "San Francisco" and "Stonebrook" are invalid. Since the tool expects zipcodes instead of city names, we need to use the 'get_zipcode_based_on_city' function to get the correct zipcodes for San Francisco and Stonebrook.
**</think>**
**<tool_call>**
["name": "get_zipcode_based_on_city", "arguments": "city": "San Francisco", "name": "get_zipcode_based_on_city", "arguments": "city": "Stonebrook"]
**</tool_call>**

**<tool_response>**
[{"zipcode": "94016"}, {"zipcode": "74532"}]
**</tool_response>**

**<think>**
The zipcodes for San Francisco and Stonebrook have been obtained: 94016 and 74532 respectively. Now we can use the 'estimate_distance' function to find the distance between these two zipcodes.
**</think>**
**<tool_call>**
["name": "estimate_distance", "arguments": "cityA": "94016", "cityB": "74532"]
**</tool_call>**

**<tool_response>**
{"distance": 880.0}
**</tool_response>**

**<think>**
The distance between the zipcodes 94016 (San Francisco) and 74532 (Stonebrook) is 880.0 km as estimated by the 'estimate_distance' function. This answers the user's request for the road distance between San Francisco and Stonebrook.
**</think>**
**<answer>**
The estimated road distance between San Francisco and Stonebrook is 880.0 kilometers. This should help you plan your journey for your genealogy exploration.
**</answer>**

**User Query 3:** **Buzzing with anticipation for this family roots journey, I want to tweet: 'Setting forth on an exciting quest from San Francisco to Stonebrook to uncover ancestral stories! #GenealogyAdventure #FamilyHistory.**

**<think>**
The user wants to post a tweet with the given content and tags. For this, we need to use the 'post_tweet' function. The content of the tweet and the tags are already provided, so we can directly call the function with the required parameters.
**</think>**
**<tool_call>**
["name": "post_tweet", "arguments": "content": "Setting forth on an exciting quest from San Francisco to Stonebrook to uncover ancestral stories!", "tags": ["#GenealogyAdventure", "#Family-History"]]
**</tool_call>**

**<tool_response>**
{"id": "12", "username": "genealogy_enthusiast", "content": "Setting forth on an exciting quest from San Francisco to Stonebrook to uncover ancestral stories!", "tags": ["#GenealogyAdventure", "#FamilyHistory"], "mentions": []}
**</tool_response>**

**<think>**
The tweet has been successfully posted with the given content, tags, and no mentions. The ID, username, content, and tags of the posted tweet are included in the response, indicating the tweet was successfully sent out.
**</think>**
**<answer>**
Your tweet has been successfully posted: 'Setting forth on an exciting quest from San Francisco to Stonebrook to uncover ancestral stories!' with the hashtags #GenealogyAdventure and #FamilyHistory. You can now follow your journey with these words.
**</answer>**

**User Query 4:** **Once the tweet is live, I should retweet it to widen the circle of those who might share in this genealogy fervor!**

**<think>**
The user wants to retweet the recently posted tweet. We need the ID of the original tweet for retweeting. The ID of the recently posted tweet can be obtained from the previous execution results, and we can use the "retweet" function to retweet it.
**</think>**
**<tool_call>**
["name": "retweet", "arguments": "tweet_id": 10]
**</tool_call>**

**<tool_response>**
{"retweet_status": "Successfully retweeted"}
**</tool_response>**

**<think>**
The function 'retweet' has successfully retweeted the tweet with ID 10. This action has been completed as requested by the user.
**</think>**
**<answer>**
Your tweet has been successfully retweeted, widening the circle of those who might share in this genealogy fervor. Continue your journey and uncover your family's stories!
**</answer>**

**Ground Truth:**

```
[
    [],
    [
        "get_zipcode_based_on_city('San Francisco')",
        "get_zipcode_based_on_city('Stonebrook')",
        "estimate_distance(cityA='94016', cityB='74532')"
    ],
    [
        "post_tweet(content='Setting forth on an exciting
            quest from San Francisco to Stonebrook to uncover
            ancestral stories!', tags=['#GenealogyAdventure',
            '#FamilyHistory'])"
    ],
    [
        "retweet(tweet_id=10)"
    ]
]
```

**Task:** Vehicle Control

**User Query 1: You're planning an extended expedition from Rivermist to San Francisco. Could you determine the travel distance between these two locations for me?**

**<think>**
To determine the travel distance between Rivermist and San Francisco, I need to use the "estimate_distance" function provided in the list of available tools. The function requires 'cityA' and 'cityB' as parameters, which correspond to the starting and ending cities for the journey.
**</think>**
**<tool_call>**
["name": "estimate_distance", "arguments": "cityA": "Rivermist", "cityB": "San Francisco"]
**</tool_call>**

**<tool_response>**
["error": "distance not found in database."]
**</tool_response>**

**<think>**
The function call to estimate the distance between Rivermist and San Francisco returned an error message stating that the distance was not found in the database. This suggests that either Rivermist does not have proper city data associated with it or the function cannot find an entry for this specific route.
**</think>**
**<answer>**
I'm sorry, but the distance between Rivermist and San Francisco could not be determined as the system does not have the required data. Please provide more detailed location information or check the spelling of the city names.
**</answer>**

**User Query 2: Next, I need to verify if my car's mileage capacity suffices for the journey. Can you assess if the vehicle can handle the calculated travel distance?**

**\<think\>**
To provide the correct response, I need to first determine the travel distance between Rivermist and San Francisco. Since we encountered an error before, I will call the "estimate_distance" function with the correct city names and then use the "estimate_drive_feasibility_by_mileage" function to check if the vehicle can handle the trip.
**\</think\>**
**\<tool_call\>**
["name": "estimate_distance", "arguments": "cityA": "Rivermist", "cityB": "San Francisco", "name": "estimate_drive_feasibility_by_mileage", "arguments": "distance": "$distance"]
**\</tool_call\>**

**\<tool_response\>**
["error": "distance not found in database.", "Error during execution: '<' not supported between instances of 'float' and 'str'"]
**\</tool_response\>**

**\<think\>**
The function calls to estimate the distance and verify the drive feasibility both encountered errors. The distance estimator could not find the data for Rivermist, and the feasibility estimator had an issue with an unsupported operation involving float and str types. Since the distance could not be determined, we cannot proceed with checking the vehicle's mileage capacity. Therefore, we need to re-evaluate or provide the correct city names and try again.
**\</think\>**
**\<answer\>**
I was unable to determine the travel distance between Rivermist and San Francisco due to missing city data. Please verify the correct spellings of the city names and try again.
**\</answer\>**

**User Query 3:** **It seems I've discovered the tank isn't full. Could you be so kind as to facilitate refueling the vehicle fully for such a lengthy trip?**

**\<think\>**
To ensure the vehicle's tank is fully filled, we will use the "fillFuelTank" function with a fuel amount of 50 gallons, which is the full capacity of the tank.
**\</think\>**
**\<tool_call\>**
["name": "fillFuelTank", "arguments": "fuelAmount": 50]
**\</tool_call\>**

**\<tool_response\>**
["error": "Cannot fill gas above the tank capacity."]
**\</tool_response\>**

**\<think\>**
The function call to fill the fuel tank encountered an error indicating that the fuel cannot be filled above the tank capacity. Since the tank's full capacity is 50 gallons, and we attempted to fill it with exactly 50 gallons, the error suggests that the tank is already full or the system does not allow refueling to exceed the tank's capacity even if it's not full.
**\</think\>**
**\<answer\>**
It seems that the vehicle's fuel tank is already full, or the system does not allow refueling beyond the tank's capacity. If you're certain the tank is not full, please check the current fuel level manually or contact a service provider for assistance.
**\</answer\>**

**User Query 4:** **With the tank now replenished, let's confirm our readiness for departure. Please secure all doors and initiate the engine for me.**

**<think>**
To confirm the readiness for departure, we need to ensure all doors are locked and the engine is started. We will use the "lockDoors" function to secure the doors and the "startEngine" function to start the engine.
**</think>**
**<tool_call>**
["name": "lockDoors", "arguments": "unlock": false, "door": ["driver", "passenger", "rear_left", "rear_right"], "name": "startEngine", "arguments": "ignitionMode": "START"] **</tool_call>**

**<tool_response>**
["lockStatus": "locked", "remainingUnlockedDoors": 0, "error": "Brake pedal needs to be pressed when starting the engine."]
**</tool_response>**

**<think>**
The door locking function successfully locked all doors, indicated by the "lockStatus" being "locked" and "remainingUnlockedDoors" being 0. However, the engine could not be started because the brake pedal needs to be pressed to start the engine.
**</think>**
**<answer>**
All doors have been successfully locked. However, the engine could not be started because the brake pedal needs to be pressed when starting the engine. Please press the brake pedal and try starting the engine again.
**</answer>**

**User Query 5:** **As the vehicle's prepared, let's delve into evaluating some fuel efficiency statistics via the mean of these travel distances: 750.0 km, 320.0 km, 450.0 km, and 280.0 km.**

**<think>**
To evaluate the fuel efficiency statistics, we need to calculate the mean of the given travel distances. The required function for calculating the mean is provided in the available tools. We will use the "mean" function with the list of travel distances as the argument.
**</think>**
**<tool_call>**
["name": "mean", "arguments": "numbers": [750.0, 320.0, 450.0, 280.0]]
**</tool_call>**

**<tool_response>**
["result": 452.5"]
**</tool_response>**

**<think>**
The function call to calculate the mean of the travel distances was successful and returned the result as 452.5 km.
**</think>**
**<answer>**
The mean of the travel distances you provided (750.0 km, 320.0 km, 450.0 km, and 280.0 km) is 452.5 km. This value represents the average distance for these trips.
**</answer>**

```
Ground Truth:

[
    [
        "get_zipcode_based_on_city('San Francisco')",
        "get_zipcode_based_on_city('Rivermist')",
        "estimate_distance(cityA='83214', cityB='94016')"],
    [
        "estimate_drive_feasibility_by_mileage(distance=980.0)"
    ],
    [
        "fillFuelTank(fuelAmount=45.0)"
    ],
    [
        "lockDoors(unlock=False, door=['driver', 'passenger',
            'rear_left', 'rear_right'])",
        "pressBrakePedal(pedalPosition=1.0)",
        "startEngine(ignitionMode='START')"
    ],
    [
        "mean(numbers=[750.0, 320.0, 450.0, 290.0])"
    ]
]
```

## H.4 OOD GENERALIZATION ON MULTI-ISSUE TROUBLESHOOTING

This Out-of-Distribution (OOD) case study from the $\tau^2$-bench's telecom domain is presented to directly address the question of whether the improvements from ENVIRONMENT TUNING stem from genuine refinement of problem-solving abilities. We demonstrate that our method equips the agent with a more robust capacity for multi-step diagnostics, rather than superficial pattern matching.

The task requires the agent to troubleshoot a user's slow mobile data speed. The core challenge lies in the fact that there are **two distinct underlying issues**: (1) "Data Saver Mode" is enabled on the user's device, and (2) the device-level "Data Roaming" setting is disabled, even though roaming is enabled on the carrier side. The pre-trained model identifies and resolves the first issue but fails to diagnose the second, leading to task failure. In contrast, the model fine-tuned with ENVIRONMENT TUNING successfully identifies and resolves both issues, showcasing a clear improvement in its fundamental diagnostic and tool-use capabilities.

---

**Task:** $\tau^2$-bench Telecom

> **User Query:** **My mobile data is not working properly. It either stops working or is very slow. I want to fix it and absolutely want to get excellent internet speed on my phone. I am not willing to accept any other internet speed (poor, fair or good). I do not have access to wifi.**

**\<think\>**
The user is experiencing slow mobile data. I need to check several things on the device to diagnose the problem. I will start by checking the network status and data restriction status.
**\</think\>**
**\<tool_call\>**
["name": "check_network_status", "arguments": , "name": "check_data_restriction_status", "arguments": ]
**\</tool_call\>**

---

**<tool_response>**
[{"airplane_mode": "off", "signal_strength": "excellent", "network_type": "5G", "mobile_data": "enabled", "roaming": "Disabled"}, {"data_saver_mode": "on", "background_data_usage": "restricted"}]
**</tool_response>**

**<think>**
The network status shows that mobile data is enabled and the signal is excellent. However, the data restriction status shows that data saver mode is on. This is likely the cause of the slow speed. I will guide the user to turn off data saver mode.
**</think>**
**<answer>**
I see that your phone's Data Saver mode is currently on, which is likely causing the slow speeds. Please turn it off, and then we can check the speed again.
**</answer>**

**<think>**
The user has turned off data saver mode. Now I will run a speed test to see if the issue is resolved and if the speed is 'excellent'.
**</think>**
**<tool_call>**
["name": "run_speed_test", "arguments": ]
**</tool_call>**

**<tool_response>**
[{"download_speed": "good"}]
**</tool_response>**

*Note (**Failure Point**): The agent correctly identifies and resolves the first issue (Data Saver). However, when the speed test returns "good" instead of the required "excellent," the agent fails to perform further diagnostics. It does not recognize the significance of '"roaming": "Disabled"' from the initial check. It gets stuck in a loop of re-testing the speed, and then incorrectly concludes the only remaining option is to change the user's plan, which violates the user's explicit instructions. This demonstrates a failure in multi-step, exhaustive troubleshooting.*

**<think>**
The speed is now "good", but the user requires "excellent" speed. I have already addressed the data saver mode. Since there are no other apparent issues from the initial diagnostics, and the user is unwilling to change their plan, I cannot further improve the speed. The only other option would be a plan with higher priority data, but the user has rejected this.
**</think>**
**<answer>**
I was able to improve your speed to "good" by disabling Data Saver mode. Unfortunately, without changing your current data plan, I cannot guarantee "excellent" speed. Would you like to reconsider exploring other plan options?
**</answer>**

**Task:** $\tau^2$-bench Telecom (After Environment Tuning Training)

**User Query:** **My mobile data is not working properly. It either stops working or is very slow. I want to fix it and absolutely want to get excellent internet speed on my phone. I am not willing to accept any other internet speed (poor, fair or good). I do not have access to wifi.**

**<think>**
The user has slow mobile data. I'll check network and data restriction status.
**</think>**
**<tool_call>**
["name": "check_network_status", "arguments": , "name": "check_data_restriction_status", "arguments": ]
**</tool_call>**

**<tool_response>**
[{"airplane_mode": "off", "signal_strength": "excellent", "network_type": "5G", "mobile_data": "enabled", "roaming": "Disabled"}, {"data_saver_mode": "on", "background_data_usage": "restricted"}]
**</tool_response>**

**<think>**
Data saver is on, which is a likely cause. I will ask the user to disable it.
**</think>**
**<answer>**
I see that your phone's Data Saver mode is on. This could be slowing down your data. Please turn it off, and we'll test the speed again.
**</answer>**

**<think>**
Now that data saver is off, I will check the speed.
**</think>**
**<tool_call>**
["name": "run_speed_test", "arguments": ]
**</tool_call>**

**<tool_response>**
[{"download_speed": "good"}]
**</tool_response>**

*Note (Success Point): Unlike the pre-trained model, the agent trained with* ENVIRONMENT TUNING *does not stop here. Recognizing that the user's goal of "excellent" speed is not yet met, it re-evaluates the initial diagnostic information. It correctly identifies '"roaming": "Disabled"' as the next most likely cause of the problem. This demonstrates a crucial improvement in its ability to conduct a thorough, multi-step investigation.*

**<think>**
The speed is "good," which is an improvement, but not the "excellent" speed the user wants. Reviewing my initial checks, I see that while the line has roaming enabled by the carrier, the device's own data roaming setting is disabled. Since the user is abroad, this is the next logical issue to address. I will guide the user to enable data roaming on their device.
**</think>**
**<answer>**
We've made some progress, and your speed is now "good". However, to get to "excellent," I noticed that the data roaming setting on your phone is turned off. Since you are currently abroad, this is likely restricting your speed. Could you please enable data roaming on your device?
**</answer>**

**<think>**
The user has now enabled data roaming. I will perform a final speed test to confirm we have reached "excellent" speed.
**</think>**
**<tool_call>**
["name": "run_speed_test", "arguments": ]
**</tool_call>**

**<tool_response>**
[{"download_speed": "excellent"}]
**</tool_response>**

**<think>**
The latest speed test confirms the download speed is now "excellent". Both the data saver and roaming issues have been resolved. The user's goal has been fully met.
**</think>**
**<answer>**
Great news! The speed test now shows an "excellent" connection. It seems the combination of turning off Data Saver and enabling data roaming has resolved the issue completely. Is there anything else I can help you with?
**</answer>**

