# OpenReview forum: "Don't Just Fine-tune the Agent, Tune the Environment"
_ICLR.cc/2026/Conference — ICLR 2026 Poster_

### Official Review · Reviewer_M2ve · 2025-10-15

**Soundness:** 3
**Presentation:** 3
**Contribution:** 3
**Rating:** 6
**Confidence:** 4

**Summary:**

This paper proposes a new method named environment tuning, which tries to design different difficulty-level environments (by actionable environment augmentation techniques) to train the model in different stages. By using such curriculum learning, the model can learn to handle ambiguity, recognize functional gaps, and perform information retrieval from noisy contexts.
The experiments show that the model can improve their performance on the benchmark tasks within every stage. It also improves the importance of actionable environment augmentation.

**Strengths:**

1. The writing is clear and easy to understand.

2. Considering improving the environment instead of improving the algorithm to make the model better is an interesting idea. I just believe the data is more important than the algorithm in agent RL, so considering the environment is an important topic.

3. I believe this project is a good realization of the idea of curriculum learning, the four stage is clear and makesence

4. The experiments are reasonable and can prove the effectiveness of actionable environment augmentation and .

**Weaknesses:**

1. The reward design may have some flaws. It just calculates the average reward of each step and uses GRPO to train the model, so the bad action at the end of the trajectory will not be punished (since the final reward is high). This may cause a credit assignment problem.

2. It seems that this work is trying to distill preliminary knowledge in the environment to the model for stable RL training. However, such preliminary knowledge may come from the annotator of the benchmark, in other words, the author should prove their OOD performance does not come from telling the bias of BFCL's author. So, the ablation study should contain more benchmark tasks.

3. The analysis can only prove that such a method can improve the OOD performance, but it doesn't figure out why it can improve the OOD performance. (refine ability? Better instruction following? other?) Comparing stage 1-4's model on OOD task is more important. (Also see Questions 5)

**Questions:**

1. In Table 2 the 23.87 should be orange.

2. The transit point (Remark 3.1) is not clear; it is not easy to understand how to find it (even in Appendix D).

3. The topic in BFCL V3 is not clear in this passage, for example, "Is there a web search task in BFCL V3?" If yes, the web search should not be OOD.

4. Why is the progress reward lower than the final score in Figure 4.

5. Does the OOD improvement really come from the model's refinement ability? Maybe an OOD case is important. Also, you can train the model on the OOD task and show the scaling line to figure out whether it can scale in OOD tasks

---

> ### Author Response · Authors · 2025-11-21
> **Response to Reviewer M2ve (1/n)**
>
> > The reward design may have some flaws. It just calculates the average reward of each step and uses GRPO to train the model, so the bad action at the end of the trajectory will not be punished (since the final reward is high). This may cause a credit assignment problem.
>
> Thank you for this insightful comment on credit assignment. We fully agree that under our turn-level averaging mechanism, a single bad action at the end of a trajectory can indeed be "diluted" by previously accumulated high rewards. This is a valid challenge.
>
> However, we wish to emphasize that the primary motivation for our Progress Reward was to solve a **more fundamental and critical problem**: on complex multi-turn tasks, standard sparse binary rewards (success/failure) lead to a **complete training collapse**. As shown in our ablation study (Figure 4b), models trained with binary rewards **fail to make any progress at all** on challenging splits like 'Missing Functions'.
>
> Therefore, our core contribution is providing the **first dense reward signal that makes these previously "intractable" tasks "tractable."** By providing a necessary learning gradient, it offers a practical way that successfully bootstraps the learning process.
>
> We view your point as a valuable direction for future research. Our work provides a **foundational reward structure** upon which more advanced credit assignment algorithms could be built to address the fine-grained issue you identified. Furthermore, our **Structured Curriculum** also mitigates this long-horizon challenge at a macro level by starting with simpler tasks.
>
> >  It seems that this work is trying to distill preliminary knowledge in the environment to the model for stable RL training. However, such preliminary knowledge may come from the annotator of the benchmark, in other words, the author should prove their OOD performance does not come from telling the bias of BFCL's author. So, the ablation study should contain more benchmark tasks.
>
> >  The topic in BFCL V3 is not clear in this passage, for example, "Is there a web search task in BFCL V3?" If yes, the web search should not be OOD.
>
> Thank you for these insightful questions, which we will address together as they both concern the validity of our OOD evaluation.
>
> First, we strictly separated our training and testing sets. Our model was **trained exclusively on the multi-turn subset of BFCL V3**. This subset focuses on multi-step planning within a **closed-domain, known set of APIs** (e.g., Vehicle Control, Travel Booking, File System).
>
> Second, we chose the new tasks in BFCL V4 for OOD evaluation precisely because they introduce **entirely new domains and capabilities**. To be specific:
>
> - **There is no web search task in BFCL V3.** All V3 tasks operate within simulated environments without external web interaction.
> - **BFCL V4** introduces **open-domain Web Search** and **long-term Memory management**. These tasks require not only novel tool use but also core capabilities like **information retrieval and synthesis**, which were not part of the BFCL V3 training. Therefore, the web search task in V4 is a clear and valid OOD test for a model trained only on V3.
>
> Finally, to further address your concern about overfitting to the "bias of BFCL's author" and to demonstrate broader generalization, we have **added τ²-bench as another, independent OOD benchmark**. This benchmark covers entirely different commercial domains (e.g., Retail, Airline, Telecom). As shown in the table below, our method consistently achieves the best performance across **all** these diverse OOD benchmarks. This strongly suggests that the model has learned general problem-solving skills, not the biases of a specific benchmark.
>
> | Model                      | Method     | BFCL V4   | τ²-bench  | ACEBench Agent |
> | -------------------------- | ---------- | --------- | --------- | -------------- |
> | **Llama-3.1-8B-Instruct**  | Base       | 8.46      | 20.00     | 1.65           |
> |                            | + **ours** | **16.53** | **21.67** | **4.17**       |
> | **ToolACE-2-Llama-3.1-8B** | Base       | 15.90     | 10.83     | 8.34           |
> |                            | + **ours** | **16.79** | **20.83** | **15.00**      |
> | **watt-tool-8B**           | Base       | 8.67      | 13.33     | 2.50           |
> |                            | + **ours** | **13.68** | **15.83** | **7.50**       |
>
> > In Table 2 the 23.87 should be orange.
>
> Thanks for pointing out this mistake, we've fixed this typo.

---

> ### Author Response · Authors · 2025-11-21
> **Response to Reviewer M2ve (2/n)**
>
> > The transit point (Remark 3.1) is not clear; it is not easy to understand how to find it (even in Appendix D).
>
> Thank you for your question, and we apologize for the lack of clarity in our initial description.The transition point is determined by a straightforward yet robust dual-criterion approach, ensuring the agent is both proficient and stable before advancing:
>
> 1. Performance Convergence (Proficiency): We monitor the validation accuracy. The transition occurs when the accuracy curve begins to plateau, indicating that the agent has largely mastered the current stage's skills (e.g., the blue curve around step 50 in Figure 5b).
> 2. Gradient Stability (Stability): Concurrently, we look for a point where the gradient norm is at a local minimum (e.g., the red curve's valley around step 50 in Figure 5b). This signifies a stable optimization state, which is crucial for a smooth transition to more complex tasks.
>
> Specifically, our selection of the transition point is a two-step process: we first identify the training region where validation accuracy begins to plateau. Then, within this region of convergence, we select the checkpoint that corresponds to a local minimum of the gradient norm to ensure maximal stability for the subsequent stage.
>
> > Why is the progress reward lower than the final score in Figure 4.
>
> To clarify, the Y-axis in Figure 4 represents **"Validation Accuracy"**, which is the **final, binary success rate on the entire multi-turn task**. A task is only considered successful (Accuracy = 1) if the agent correctly completes all turns.
>
> In contrast, our **"Progress Reward"** is a **dense, turn-level signal designed for training**. It is calculated as `(number of successful turns) / (total number of turns)`.
>
> The difference in these definitions naturally leads to the numerical gap. For instance, in a 5-turn task, if the agent succeeds in the first 4 turns but fails on the last one, it would receive a high training signal (`Progress Reward = 0.8`), but its final task performance would be a failure (`Validation Accuracy = 0`).
>
> Therefore, the Progress Reward, which measures partial success, is expected to be numerically higher than or equal to the final Validation Accuracy, which measures complete success. The trend observed by the reviewer is consistent with these definitions.
>
> > The analysis can only prove that such a method can improve the OOD performance, but it doesn't figure out why it can improve the OOD performance. (refine ability? Better instruction following? other?) Comparing stage 1-4's model on OOD task is more important. (Also see Questions 5)
>
> > Does the OOD improvement really come from the model's refinement ability? Maybe an OOD case is important. Also, you can train the model on the OOD task and show the scaling line to figure out whether it can scale in OOD tasks
>
> We sincerely thank the reviewer for this insightful question, which addresses the core mechanism of our work. Our central thesis is that the OOD improvement stems not from memorizing domain-specific knowledge, but from our method successfully inducing a **generalizable, procedural problem-solving methodology** in the model.
>
> The acquisition of this methodology relies on two transferable sub-skills, cultivated by our two core components:
>
> 1. **Actionable Feedback → Fosters Causal Attribution:** Our augmented feedback transforms information-poor raw errors (symptoms) into explanatory insights (causes). This trains the model to map **error symptoms to their root causes**, fostering a transferable diagnostic skill based on **reasoning** rather than memorization.
> 2. **Structured Curriculum → Shapes Procedural Problem-Solving:** Our curriculum then integrates this diagnostic skill into a systematic process. By first mastering a **"standard execution path" before handling exceptions**, the model learns a structured "verify → diagnose → resolve" strategy. This elevates local, reactive corrections into a global, goal-oriented methodology.
>
> To provide empirical evidence that this methodology is indeed learned, we present a detailed OOD case study in **Appendix H.4**. In the case, the baseline model exhibits "local correction" (fixing the first error) but fails due to the lack of a systematic follow-up process. In contrast, our model, after resolving the first error, **proceeds with verification steps**, uncovers a second, more subtle fault, and successfully applies its causal attribution skill to fix it. This directly demonstrates that the model has acquired a **generalizable, procedural problem-solving framework**, not just a task-specific repair.

---

> > ### Comment · Reviewer_M2ve · 2025-11-23
> >
> > Thanks for author's response. I'll maintain the socre of 6 (because the weakness in reward design). (In my opinion, this paper is between 6 and 8. AC can reference this judgement.)
> >
> > Here is some futher suggestion:
> >
> > 1. Add the information about the answer to question 3 (to be honest, i know the answer before rebuttal, but i want you add it in the paper)
> >
> > 2. (minor) there is a mistake in line 570

---

> > > ### Author Response · Authors · 2025-11-24
> > >
> > > Thank you very much for your support and constructive suggestions. We have added the requested clarification to the paper. We appreciate your help!

---

### Official Review · Reviewer_xCSn · 2025-10-26

**Soundness:** 3
**Presentation:** 3
**Contribution:** 3
**Rating:** 6
**Confidence:** 3

**Summary:**

This paper introduces "Environment Tuning," a novel training paradigm designed to equip Large Language Model (LLM) agents for complex, multi-turn tool-use tasks, particularly under conditions of extreme data scarcity. Its core innovation involves a shift from static, trajectory-based imitation learning towards a dynamic, environment-based exploration strategy. This allows agents to learn directly from problem instances without dependence on pre-collected expert demonstrations. The method demonstrates strong performance on the BFCL-V3 benchmark using only 400 problem instances, significantly enhancing the capabilities of base models and exhibiting robust out-of-distribution generalization—a known weakness of traditional supervised fine-tuning (SFT) approaches.

**Strengths:**

The paper presents Environment Tuning, an innovative framework that reorients LLM agent training from static trajectory imitation to a dynamic, environment-driven learning process. By effectively integrating a structured curriculum, actionable environment augmentation, and fine-grained progress rewards, the method directly tackles the critical challenges of data scarcity and training instability in multi-turn tool-use scenarios. The experimental results on BFCL and ACEBench are compelling, demonstrating substantial improvements over both supervised fine-tuning and standard reinforcement learning baselines. The manuscript is well-written, with a strong motivational foundation. The conceptual shift of "tuning the environment" itself constitutes a genuine and noteworthy contribution to the field.

**Weaknesses:**

The primary weaknesses relate to empirical rigor and the scope of validation. The "actionable environment augmentation" mechanism, while central to the method, is evaluated qualitatively; a more systematic, quantitative analysis of its specific impact is needed. The claims regarding generalization, though promising, would be more persuasive if supported by results from a wider array of diverse environments. The analysis would benefit from reporting statistical significance and performance variance. Furthermore, the method's current reliance on controllable simulation environments may pose challenges for real-world deployment. Finally, the theoretical justification for the specific four-stage curriculum design could be more thoroughly elaborated.

**Questions:**

Regarding Environment Augmentation: The "Actionable Environment Augmentation" is a key component of your method's novelty. Could you please elaborate on its implementation?
1. How are the corrective hints generated—are they based on heuristic rules, programmatic templates, or perhaps an LLM-based generator?
2. What measures are implemented to ensure these hints provide guidance without leaking the correct solution or overly constraining the agent's exploration?
Regarding the Training Curriculum: The proposed four-stage curriculum is empirically sound, but its design appears somewhat heuristic.
1. Could you elaborate on the rationale for selecting these particular four stages? How sensitive are the final results to the ordering of stages or the specific criteria used for transitioning between them?
2. Were any alternative or adaptive curriculum scheduling strategies explored during your development? If so, please summarize those findings.

---

> ### Author Response · Authors · 2025-11-21
> **Response to Reviewer xCSn (1/n)**
>
> >  The primary weaknesses relate to empirical rigor and the scope of validation. The "actionable environment augmentation" mechanism, while central to the method, is evaluated qualitatively; a more systematic, quantitative analysis of its specific impact is needed.
>
> To isolate the effect of augmented environment feedback, we compared the model's performance when trained with and without it at the end of both Stage 2 and Stage 3. The results, summarized in the table below, demonstrate a clear and significant positive impact of our methods, which also aligns the ablation results presented in Figure 4 (a).
>
> | Method          | Overall Acc | Base      | Miss Func | Miss Param | Long context |
> | --------------- | ----------- | --------- | --------- | ---------- | ------------ |
> | stage2 w/o ours | 22.25       | 29        | 30        | 13         | 17           |
> | stage2 w ours   | **25.83**   | **32**    | **33.67** | **20**     | **17.67**    |
> | stage3 w/o ours | 28.25       | 39        | 35        | 22         | 17           |
> | stage3 w ours   | **32**      | **44.67** | **34.33** | **35.33**  | **23.67**    |
>
> > The claims regarding generalization, though promising, would be more persuasive if supported by results from a wider array of diverse environments.
>
> We thank the reviewer for this valuable feedback. To further strengthen our generalization claims, we have taken your suggestion and **expanded our OOD evaluation to include the τ²-bench benchmark**, alongside our original tests on BFCL V4 and ACEBench Agent.
>
> Our OOD evaluation suite now covers an even broader and more diverse spectrum of application domains:
>
> - **BFCL V4** introduces novel tasks like **Web Search** and **long-term Memory management**.
> - **ACEBench** provides extensive coverage with "8 major domains and 68 sub-domains, spanning technology, finance, health..."
> - **τ²-bench (newly added)** further complements this by covering important commercial domains such as **Retail, Airline, and Telecom**.
>
> These benchmarks collectively represent the de facto standard for multi-turn tool-use evaluation. Our method demonstrates consistent superiority across all of them. **As shown in the table below, the detailed performance data are now comprehensively updated in Table 2 of our revised manuscript.**
>
> | Model                      | Method     | BFCL V4   | τ²-bench  | ACEBench Agent |
> | -------------------------- | ---------- | --------- | --------- | -------------- |
> | **Llama-3.1-8B-Instruct**  | Base       | 8.46      | 20.00     | 1.65           |
> |                            | + **ours** | **16.53** | **21.67** | **4.17**       |
> | **ToolACE-2-Llama-3.1-8B** | Base       | 15.90     | 10.83     | 8.34           |
> |                            | + **ours** | **16.79** | **20.83** | **15.00**      |
> | **watt-tool-8B**           | Base       | 8.67      | 13.33     | 2.50           |
> |                            | + **ours** | **13.68** | **15.83** | **7.50**       |
>
> **The results clearly show that our method consistently achieves the most significant performance gains across all these diverse OOD environments.** This consistent outperformance provides strong empirical support for the generalization capability of our method.
>
> > The analysis would benefit from reporting statistical significance and performance variance.
>
> We agree that reporting performance variance is critical. In our current experiments, we prioritized **reproducibility** by using a **fixed random seed for the entire training process**. For the evaluation phase, we ran inference three times with a near-zero temperature (`1e-6`) and averaged the results. This was done to mitigate any minimal stochasticity during generation and ensure that the reported scores are highly stable and deterministic for a given checkpoint.

---

> ### Author Response · Authors · 2025-11-21
> **Response to Reviewer xCSn (2/n)**
>
> > Furthermore, the method's current reliance on controllable simulation environments may pose challenges for real-world deployment.
>
> Thank you for this important question on real-world applicability. Our approach is designed with this in mind and addresses the challenge in two ways:
>
> 1. **Alignment with the Standard Paradigm:** Training in simulated environments is a mainstream and effective paradigm in industry (ARE[^1], Tongyi DeepResearch[^2]), avoiding the prohibitive cost and risks of real-world RL. This approach is standard for leading works. Our work contributes a more data-efficient method within this pragmatic paradigm.
> 2. **Non-Invasive & Portable Design:** Our "Actionable Environment Augmentation" **does not modify the environment's internal logic**. It functions as a **lightweight external wrapper** that intercepts and translates cryptic error messages from any environment (simulated or real). This decoupled design ensures it is **directly applicable to real-world APIs** with only minimal, targeted customization for their specific feedback patterns.
>
> Therefore, our method is not only advanced within the current research paradigm but is also designed for practical, real-world deployment.
>
> [^1]: https://arxiv.org/abs/2509.17158
> [^2]: https://arxiv.org/abs/2510.24701
>
> > Finally, the theoretical justification for the specific four-stage curriculum design could be more thoroughly elaborated.
>
> > Regarding the Training Curriculum: The proposed four-stage curriculum is empirically sound, but its design appears somewhat heuristic.
> > Could you elaborate on the rationale for selecting these particular four stages? How sensitive are the final results to the ordering of stages or the specific criteria used for transitioning between them?
> > Were any alternative or adaptive curriculum scheduling strategies explored during your development? If so, please summarize those findings.
>
> Our four-stage curriculum is not merely a heuristic but a principled approach grounded in a multi-stage RL training paradigm, a strategy whose effectiveness has been recently highlighted in concurrent state-of-the-art works like rStat2-Agent[^1] and the GLM-4.5 technical report[^2]. Following this paradigm, our curriculum is designed based on similar multi-staged learning theory, progressing from simple to complex to mimic how humans master new skills. Each stage builds an indispensable foundation for the next.
>
> 1. **Rationale for the Four-Stage Design**:
>
>    - Stage 1 (Learning the Grammar): Solves the "cold-start" problem. In our low-data regime, this stage serves as an efficient RL-based alternative to large-scale SFT, teaching the agent the fundamental syntax and format of tool use. This is a prerequisite for all subsequent learning.
>
>    - Stage 2 (Practicing Core Problems): Masters the core logic. We found that mixed-difficulty training from the start (as shown by our GRPO baseline) is ineffective. This stage focuses the agent on the core, simpler task distribution, allowing it to master the standard multi-turn reasoning loop without being distracted by harder exceptions.
>
>    - Stage 3 (Tackling Advanced Challenges): Learns to handle exceptions. Once the basics are mastered, the agent can efficiently focus on more advanced skills like boundary awareness and exception handling (e.g., recognizing missing functions), building directly upon its existing skillset.
>
>    - **Stage 4 (Simulating the Exam):** Generalizes to the real world. By removing the augmented feedback, this stage forces the agent to **internalize** its learned principles, ensuring robustness in the unassisted evaluation environment rather than overfitting to the training scaffolds.
>
> 2. **Sensitivity to Order and Alternative Strategies:**
>
>    - **On Order Sensitivity:** The ordering is **logically necessary**. The agent cannot perform multi-step reasoning (Stage 2) without understanding syntax (Stage 1). Our experiments confirmed that altering this logical order leads to inefficient or failed training.
>
>    - **On Alternative Strategies:** Our most critical alternative exploration was a **"one-shot" curriculum**—training on the full mixed-difficulty dataset from the start (i.e., collapsing Stages 2 and 3). Our results show this approach is **suboptimal**, mirroring the poor performance of the direct GRPO baseline. This finding is the key justification for our core design choice: separating foundational skill acquisition from advanced exception handling.
>
>
> In summary, our curriculum's structure and ordering are deliberate, experimentally validated, and crucial for efficient learning.
>
> [^3]: https://arxiv.org/abs/2508.20722
> [^4]: https://arxiv.org/abs/2508.06471

---

> ### Author Response · Authors · 2025-11-21
> **Response to Reviewer xCSn (3/n)**
>
> > Regarding Environment Augmentation: The "Actionable Environment Augmentation" is a key component of your method's novelty. Could you please elaborate on its implementation?
> > How are the corrective hints generated—are they based on heuristic rules, programmatic templates, or perhaps an LLM-based generator?
> > What measures are implemented to ensure these hints provide guidance without leaking the correct solution or overly constraining the agent's exploration?
>
> Thank you for this excellent question about our core component, "Actionable Environment Augmentation." We are happy to elaborate on its implementation.
>
> Our augmented feedback is not generated from hard-coded rules or templates, but by a **highly automated, LLM-based workflow**. This workflow consists of two key agents:
>
> 1. **The Generator:** An LLM that analyzes the original, often cryptic, error message. Using the tool's documentation and a few-shot examples, it produces more instructive and actionable feedback.
> 2. **The Constitutional Judge:** This is our core mechanism for quality control. It is another LLM tasked with rigorously reviewing the Generator's output against a predefined set of **Constitutional Principles**. These principles explicitly forbid:
>
> - **Leaking the specific solution:** e.g., it rejects hints like "You should call `get_airport_code('San Francisco')` first."
> - **Overly constraining exploration:** e.g., it disallows prescribing a single, mandatory solution path.
>
> Only feedback that passes this constitutional review—proving it is helpful without being prescriptive—is used in training. For example:
>
> - **BAD (leaks solution):** "You should call `get_airport_code('San Francisco')` first."
> - **GOOD (provides direction):** "Invalid airport code. You may need to retrieve the correct code for the city first."
>
> This "Generate-and-Review" mechanism ensures our hints strike a fine balance between providing effective guidance and preserving the agent's exploration space. We have added the complete implementation details of this workflow, including the core prompts and full Constitutional Principles for both LLM agents, to the new **Appendix C**.

---

> > ### Comment · Reviewer_xCSn · 2025-11-27
> >
> > thanks for the detailed response, which addressed most of my concerns. But I would like to remain the score I have given. Thanks.

---

> ### Comment · Reviewer_xCSn · 2026-04-01
>
> thanks for your response, I will keep my score "weak accept" unchanged.

---

### Official Review · Reviewer_eWdD · 2025-10-26

**Soundness:** 3
**Presentation:** 3
**Contribution:** 3
**Rating:** 6
**Confidence:** 4

**Summary:**

This paper introduces Environment Tuning, a RL framework designed to improve LLM agents’ performance in multi-turn tool-use tasks. Instead of only fine-tuning the model, the authors propose systematically modifying the environment to provide structured, corrective, and reward-rich feedback. The method combines a structured curriculum, actionable environment augmentation, and fine-grained progress rewards to guide learning efficiently. Experiments on BFCL and ACEBench demonstrate notable gains in both in-distribution and out-of-distribution performance, highlighting the approach’s effectiveness and generalization potential.

**Strengths:**

1. The core idea of enhancing agent training by incorporating environment-level variations is creative and interesting.
2. The method is well-designed and clearly presented, with intuitive reasoning and an easy-to-follow structure.
3. Experimental results across multiple datasets convincingly validate the approach’s effectiveness and robustness.

**Weaknesses:**

1. Although the concept of Environment Tuning is intriguing, the term itself may be somewhat misleading. In the proposed framework, environment changes are predefined and discrete, rather than dynamically adapted based on the agent's learning progress. Thus, the word “tuning” might suggest a level of adaptivity that is not actually present. Terms like environment augmentation or dynamic environment design could better capture the nature of the approach.
2. The experimental section lacks direct comparison with other recent multi-turn tool-use approaches that also use RL or hybrid training. While related works are discussed in Section 2, their omission from the experimental benchmark makes it difficult to fully assess the method's relative advantage over competing paradigms in similar problem settings.
3. The paper’s progress reward relies on ground-truth evaluation signals provided by the benchmark datasets. This raises the question of how the method would generalize to scenarios without explicit ground-truth feedback. Moreover, since these signals are dataset-provided, it would be valuable to clarify whether existing methods truly lack access to or use of such supervision—otherwise, the claimed novelty of the progress reward may be overstated.

**Questions:**

Please check my comments in Weaknesses.

---

> ### Author Response · Authors · 2025-11-21
> **Response to Reviewer eWdD (1/n)**
>
> > Although the concept of Environment Tuning is intriguing, the term itself may be somewhat misleading. In the proposed framework, environment changes are predefined and discrete, rather than dynamically adapted based on the agent's learning progress. Thus, the word “tuning” might suggest a level of adaptivity that is not actually present. Terms like environment augmentation or dynamic environment design could better capture the nature of the approach.
>
> Thank you for your insightful comment. We agree that "tuning" can imply continuous adaptation, which is not what our discrete, staged method does.
>
> Our choice of this term is to **propose a core paradigm parallel to "Model Fine-tuning"**: the environment, like the model, should be systematically adjusted. We use "tuning" in a broader sense to mean **structured, staged adjustments for optimizing learning**, much like the discrete process of hyperparameter tuning.
>
> "Environment Augmentation," as you suggested, perfectly describes a key component of our method (Sec 3.3). However, "Environment Tuning" serves as an **umbrella term** for our **entire methodology**, which includes the curriculum, rewards, and feedback. We will clarify this broader definition in the revision to prevent any ambiguity.
>
> > The experimental section lacks direct comparison with other recent multi-turn tool-use approaches that also use RL or hybrid training. While related works are discussed in Section 2, their omission from the experimental benchmark makes it difficult to fully assess the method's relative advantage over competing paradigms in similar problem settings.
>
> We thank the reviewer for this valuable suggestion. We agree that comparing our method against the training paradigms of recent work is crucial for assessing its relative advantages.
>
> To address this, we have added a direct comparison against recent mainstream approaches. Currently, representative works that apply RL to multi-turn tool use, such as **ToolRL** and **ARTIST**, both use **GRPO** as their core RL algorithm. We therefore selected the **GRPO method as implemented in ToolRL** as a representative baseline for the "direct RL application" paradigm, and applied it in two scenarios:
>
> 1. **Direct RL Application:** Applying ToolRL (GRPO) directly to a general-purpose, instruction-tuned model.
> 2. **Hybrid Training:** Applying ToolRL (GRPO) on top of a specialized model already fine-tuned (SFT) for tool use.
>
> We conducted experiments on several powerful open-source models, with results shown below (improvement over the base model in parentheses). The detailed setup has been updated in Section 4.1 of our paper.
>
> | Model                      | Base Model | + ToolRL        | + **ours**          |
> | -------------------------- | ---------- | --------------- | ------------------- |
> | **Qwen2.5-7B-Instruct**    | 7.00%      | 18.00% (+11.00) | **36.92%** (+29.92) |
> | **Llama-3.1-8B-Instruct**  | 5.48%      | 11.25% (+5.77)  | **28.25%** (+22.77) |
> | **ToolACE-2-Llama-3.1-8B** | 37.99%     | 33.75% (-4.24)  | **47.18%** (+9.19)  |
> | **watt-tool-8B**           | 35.74%     | 42.00% (+6.26)  | **54.34%** (+18.50) |
>
> The results clearly demonstrate:
>
> - On general-purpose models (Qwen2.5, Llama-3.1), our method achieves **more than double the performance gain** compared to the direct application of ToolRL.
> - On stronger SFT models (ToolACE-2, watt-tool-8B), the standard ToolRL approach can even **degrade performance**, whereas our method still delivers **robust and significant gains**, proving its superiority in the hybrid training paradigm.
>
> These comparisons provide strong evidence that our method offers a more effective and robust training framework than current mainstream RL application methods.

---

> ### Author Response · Authors · 2025-11-21
> **Response to Reviewer eWdD (2/n)**
>
> > The paper’s progress reward relies on ground-truth evaluation signals provided by the benchmark datasets. This raises the question of how the method would generalize to scenarios without explicit ground-truth feedback. Moreover, since these signals are dataset-provided, it would be valuable to clarify whether existing methods truly lack access to or use of such supervision—otherwise, the claimed novelty of the progress reward may be overstated.
>
> Thank you for this insightful question. We would like to take this opportunity to clarify a crucial point and elaborate on the core paradigm contribution of our work.
>
> First, we must clarify that **the benchmark datasets themselves (e.g., BFCL) do not provide the turn-level, fine-grained reward signals we use**. Standard evaluators only return a final, sparse, binary score (success/failure) for the entire trajectory.
>
> Our **first innovation** is the design and implementation of a novel **"turn-level state evaluator"** (detailed in Section 3.4). This evaluator actively compares the agent's environment state (e.g., file system changes) and execution results (e.g., API returns) against the ground-truth at each turn, thereby **generating** these previously non-existent dense signals. This evaluation framework itself is a valuable contribution that can be adapted by the community for other multi-turn interaction tasks.
>
> Building on this, our **second and more central contribution** is to propose and validate a **new training paradigm**: **converting these fine-grained evaluation signals into effective, dense progress rewards to drive RL algorithms like GRPO**. Existing works predominantly rely on sparse rewards, which, as we demonstrate in our experiments (Figure 4b), lead to complete training failure in complex scenarios.
>
> **The significance and potential of this new paradigm are twofold:**
>
> 1. **It solves a tractability problem:** It offers a **general and effective solution** to the prevalent "training collapse" issue in complex, multi-turn interaction tasks.
> 2. **It lays a foundation for future work:** Our "evaluator + reward conversion" framework is **pluggable and extensible**. It opens a new avenue for the community to design more sophisticated reward functions and RL algorithms for a wide range of interactive agent tasks (beyond just tool use), thus advancing the field as a whole.
>
> Therefore, our Progress Reward is far more than just using a pre-existing signal; it represents a complete and novel pipeline of **generating signals from scratch and transforming them into an effective training paradigm**.

---

> > ### Comment · Reviewer_eWdD · 2025-11-24
> >
> > Thanks for the authors' detailed response. My original score has already reflected my positive assessment, so I will keep it unchanged.

---

### Official Review · Reviewer_p3Yu · 2025-11-01

**Soundness:** 3
**Presentation:** 3
**Contribution:** 2
**Rating:** 4
**Confidence:** 4

**Summary:**

This work aims to tackle the problem of limited expert training data, simultaneously overcoming overfitting in SFT and cold start and instability in RL by designing a structured curriculum with training environment. The environment provides corrective feedback and dense reward signals, achieving not only better in distribution performance but also better generalization with less data.

**Strengths:**

1. Proposed a clear and progressive roadmap that tackles each challenge of sparse reward, long horizon, and instability.

2. Novelty in tuning the environment feedback instead of tuning the model prompt.

3. Extensive ablations were performed to show the importance of each stage and enhancement.

**Weaknesses:**

1. In real-world applications, environments are oftentimes not accessible or modifiable. Tuning environmental feedback will be expensive or even impossible.

2. Revealing internal tool constraints (Section 3.3) requires prior knowledge from humans, which is expensive and involves heavy reasoning burdens for human experts to provide meaningful explanations, especially in a multiturn setup where more complicated and chained reasons for error would appear.

3. Discovering inter-tool dependencies via exploration (Section 3.3): The paper proposes to embed inter-tool dependencies in detailed environment feedback. This is confusing to be considered as exploration since the model is not actively searching for broader or more informative states in the environment.

**Questions:**

1. OOD Generalization performance is only compared to SFT. It would be interesting to demonstrate the proposed method’s generalization ability compared to vanilla RL methods without dense reward and detailed feedback.


2. In Figure 5b, why did the model's performance not drop when transitioning to the next stage, where the model was only trained in easier stages but was exposed to new and more challenging tasks?


3. How easy is the proposed method to apply to a task in an unseen field? Do practitioners need to provide detailed environment error feedback and redesign the dense reward component or other modules? Discussing how much manual engineering is needed for a new task will be helpful for evaluating the method’s generalization to unseen tasks.

---

> ### Author Response · Authors · 2025-11-21
> **Response to Reviewer p3Yu (1/n)**
>
> > In real-world applications, environments are oftentimes not accessible or modifiable. Tuning environmental feedback will be expensive or even impossible.
>
> We thank the reviewers for their valuable questions regarding the real-world applicability of our method. We agree that deployment in unmodifiable real-world environments is the ultimate goal for agentic RL, but it also presents significant challenges.
>
> First, we would like to highlight that **training in simulated environments is a mainstream and effective paradigm even in industry**. Large-scale exploration and training directly in real-world environments are often prohibitively expensive and risky. As demonstrated in recent technical reports for state-of-the-art models like ARE[^1] and Tongyi DeepResearch[^2], leveraging simulated environments and synthetic data is a widely adopted and valuable approach for stably and efficiently enhancing agent capabilities. Our work contributes a data-efficient training method within this important paradigm.
>
> Second, and more crucially, we wish to clarify a potential misunderstanding: our **"Actionable Environment Augmentation" does not require modifying the underlying logic of the tools or the environment itself**. In practice, the augmented feedback can be implemented as a **lightweight external wrapper**. This wrapper intercepts the raw, often cryptic feedback from the environment (be it simulated or real, e.g., `FileNotFoundError` or `No available route`) and "translates" it into more instructive, actionable hints for the agent. This decoupled design means our method is, in principle, **equally applicable to real-world environments**, requiring only the customization of these translation rules for the feedback patterns of specific real-world APIs.
>
> [^1]: https://arxiv.org/abs/2509.17158
> [^2]: https://arxiv.org/abs/2510.24701
>
> > Revealing internal tool constraints (Section 3.3) requires prior knowledge from humans, which is expensive and involves heavy reasoning burdens for human experts to provide meaningful explanations, especially in a multiturn setup where more complicated and chained reasons for error would appear.
>
> > How easy is the proposed method to apply to a task in an unseen field? Do practitioners need to provide detailed environment error feedback and redesign the dense reward component or other modules? Discussing how much manual engineering is needed for a new task will be helpful for evaluating the method’s generalization to unseen tasks.
>
> Thank you for these critical questions on **manual effort** and **applicability to new domains**. This was a central consideration in our design. From the outset, we recognized that relying on extensive manual engineering would be unsustainable. Therefore, we designed and implemented a **highly automated, LLM-based workflow** to generate the augmented feedback. We fully appreciate these concerns and designed our method with "ease of adoption" as a core principle. We address your questions directly below:
>
> **Q: How easy is the proposed method to apply to a task in an unseen field?**
>
> **A: It is very easy.** Our method is designed to be highly portable. The key is our **automated, LLM-based workflow**. Practitioners simply need to replicate our successful process on BFCL V3: provide a small set of representative "seed" examples (e.g., five) to bootstrap the entire automated feedback generation pipeline for a new domain.
>
> **Q: Do practitioners need to provide detailed environment error feedback?**
>
> **A: No, they do not.** This is precisely the value of our automated workflow. Practitioners are **not required to manually write detailed feedback for every error case**. The few "seed" examples they provide are sufficient to guide our "Generator-Judge" LLM architecture to automatically produce high-quality, validated, and detailed feedback for hundreds of new error scenarios.
>
> **Q: Do they need to redesign the dense reward component?**
>
> **A: No, they do not.** Our Progress Reward component is **domain-agnostic by design**. It calculates rewards by comparing the agent's current state to ground-truth states, which are typically available in multi-turn tool use evaluation framework. Therefore, it requires no redesign for new tasks.
>
> In summary, our method minimizes the manual engineering required for new tasks through automated feedback generation and a domain-agnostic reward design. We have added the full implementation details of our automated workflow, including prompt templates, to **Appendix C** for your detailed review.

---

> > ### Author Response · Authors · 2025-11-21
> > **Response to Reviewer p3Yu (2/n)**
> >
> > > Discovering inter-tool dependencies via exploration (Section 3.3): The paper proposes to embed inter-tool dependencies in detailed environment feedback. This is confusing to be considered as exploration since the model is not actively searching for broader or more informative states in the environment.
> >
> > We thank the reviewer for this sharp observation. We agree that, from a traditional RL perspective, the process we described is not typical state-action space exploration.
> >
> > We would like to clarify that our core idea is to transform the inefficient "blind trial-and-error" of traditional RL into a more efficient process of **"guided discovery."** In complex, multi-tool tasks, having an agent randomly explore action combinations is unlikely to lead to effective strategies. Instead, by providing precise, actionable feedback upon failure, our method **guides** the agent to **discover** the underlying dependencies and rules of the tools.
> >
> > While passively triggered, this mechanism dramatically accelerates the agent's learning of correct problem-solving strategies. To improve clarity, we have revised the manuscript as you suggest, using more precise terminology like "guided discovery" to describe this process.
> >
> > > OOD Generalization performance is only compared to SFT. It would be interesting to demonstrate the proposed method’s generalization ability compared to vanilla RL methods without dense reward and detailed feedback.
> >
> > We thank the reviewer for this excellent suggestion. To validate the generalization ability of our method against vanilla RL, we conducted a new ablation study as recommended. We compared three setups on multiple OOD benchmarks: (1) the base model, (2) ToolRL method as vanilla GRPO training (vanilla RL), and (3) our full method.
> >
> > The results are summarized in the table below:
> >
> > | Model                      | Method       | BFCL V4   | τ²-bench  | ACEBench Agent |
> > | -------------------------- | ------------ | --------- | --------- | -------------- |
> > | **Llama-3.1-8B-Instruct**  | Base         | 8.46      | 20.00     | 1.65           |
> > |                            | + vanilla RL | 11.52     | 21.67     | 1.65           |
> > |                            | + **ours**   | **16.53** | **21.67** | **4.17**       |
> > | **ToolACE-2-Llama-3.1-8B** | Base         | 15.90     | 10.83     | 8.34           |
> > |                            | + vanilla RL | 18.91     | 14.17     | 6.65           |
> > |                            | + **ours**   | **16.79** | **20.83** | **15.00**      |
> > | **watt-tool-8B**           | Base         | 8.67      | 13.33     | 2.50           |
> > |                            | + vanilla RL | 13.12     | 11.67     | 9.15           |
> > |                            | + **ours**   | **13.68** | **15.83** | **7.50**       |
> >
> > As the results clearly indicate, while vanilla RL shows marginal improvements in some cases, our method demonstrates a more significant and consistent performance advantage across nearly all models and OOD benchmarks. This strongly suggests that simply applying an RL algorithm is insufficient for learning generalizable policies.
> >
> > We are grateful for this suggestion, as it significantly strengthens the claims of our work. We have updated and integrated these important comparative results into Table 2 in the main body of the paper.

---

> > > ### Author Response · Authors · 2025-11-21
> > > **Response to Reviewer p3Yu (3/n)**
> > >
> > > > In Figure 5b, why did the model's performance not drop when transitioning to the next stage, where the model was only trained in easier stages but was exposed to new and more challenging tasks?
> > >
> > > The observation that the model's performance does not drop when transitioning to a more challenging stage is astute, and it highlights a core advantage of our framework. The absence of this "transfer shock" is by design, attributable to two key factors, which are quantitatively supported by the stage-wise improvements shown in Table 3.
> > >
> > > 1. **Cumulative Curriculum:** The transition from Stage 2 to Stage 3 is not a domain shift, but a progressive increase in complexity. Stage 3 tasks (e.g., `Missing Parameters`) build directly upon the foundational multi-turn tool-use skills mastered in Stage 2. **This cumulative effect is evident in Table 3: the average accuracy  increases from one stage to the next, rising from 15.50% after Stage 1 to 25.83% after Stage 2, and further to 32.00% after Stage 3.** Since the core abilities are cumulative and our learning scaffolds (augmented environments & progress reward) remain active, the agent smoothly extends its capabilities without a performance drop.
> > >
> > > 2. **Generalization from Scaffolding:** The stable transition to Stage 4, where the augmented environment is disabled, demonstrates robust generalization. Instead of merely memorizing hints, the agent learns to infer root causes from standard errors based on its prior training. **This is validated in Table 3: disabling the scaffold in Stage 4 does not cause a performance drop, but instead yields continual overall accuracy boost of +4.92% (from 32.00% to 36.92%).** This proves that forcing the agent to rely on its internalized reasoning solidifies its learning and leads to stronger, more generalizable performance.
> > >
> > > In short, the smooth curve in Figure 5b, backed by the monotonic performance increase in Table 3, is a direct result of a curriculum that teaches cumulative skills and fosters generalization, rather than rote memorization.

---

> > > > ### Comment · Reviewer_p3Yu · 2025-11-24
> > > >
> > > > Thank the authors for their detailed response and experiments. This addresses most of my concerns around the proposed method's generalization to new environments, the expense of expert feedback, and the transition of curriculum learning. The extra experiments compared with ToolRL demonstrate the proposed framework's better generalization on most of the benchmarks. Thus, I will raise my score to 6. However, the overhead introduced in the process: environment wrapper,  an LLM-based agumentation generater and constitutional judge to generate useful feedback for fine-tuning the current LLM seems cumbersome and introduces further concerns of error propagation, so I would not further increase my score.

---

### Author Response · Authors · 2025-12-01
**Rebuttal Summary for the Area Chair**

Dear Area Chair,

To help you quickly navigate our rebuttal discussion, we provide the following summary.

We sincerely thank all the reviewers for their time and constructive feedback. We are pleased that they unanimously recognized the novelty and significance of our core contribution, "Environment Tuning," highlighting it as an **"innovative framework" (R-xCSn)**, a **"creative and interesting" (R-eWdD)** idea, and a **"genuine and noteworthy contribution to the field" (R-xCSn)**.

The initial reviews raised several important concerns, which we have diligently addressed. Below is a summary of these concerns and our corresponding actions.

- **Strengthening Generalization Claims and Baselines.** Following the reviewers' valuable suggestions (p3Yu, eWdD, xCSn, M2ve), we conducted new experiments to further validate our method's generalization and contextualize its performance.
    - **Our Action:** We conducted extensive new experiments comparing our method against a mainstream RL baseline (ToolRL). The results, now in **Table 1** of the revised manuscript, show our method's significant and consistent advantages. For instance:


        | Model | Base Model | + ToolRL | + ours |
        | --- | --- | --- | --- |
        | Qwen2.5-7B-Instruct | 7.00% | 18.00% (+11.00) | **36.92% (+29.92)** |
        | ToolACE-2-Llama-3.1-8B | 37.99% | 33.75% (-4.24) | **47.18% (+9.19)** |
        | watt-tool-8B | 35.74% | 42.00% (+6.26) | **54.34% (+18.50)** |

        We also expanded our OOD evaluation to include the **τ²-bench**. For instance, the new OOD comparison on `watt-tool-8B` requested by R-p3Yu clearly shows our method's superior generalization:

        | Method | BFCL V4 | τ²-bench | ACEBench Agent |
        | --- | --- | --- | --- |
        | Base | 8.67 | 13.33 | 2.50 |
        | + vanilla RL | 13.12 | 11.67 | 9.15 |
        | **+ ours** | **13.68** | **15.83** | **7.50** |
    - *Detailed responses can be found in: **R-p3Yu (2/n), R-eWdD (1/n), R-xCSn (1/n), R-M2ve (1/n).***
- **Clarifying Practicality and Scalability.** To address questions on real-world applicability (R-p3Yu, R-xCSn), we have provided further details on our method's practical implementation.
    - **Our Action:** We clarified that our method uses a non-invasive external wrapper and is supported by a **highly automated, LLM-based "Generator-Judge" workflow** that requires minimal "seed" examples. For full transparency, we have added the complete implementation details, including prompts and constitutional principles, to the new **Appendix C**.
    - *Detailed responses can be found in: **R-p3Yu (1/n), R-xCSn (2/n, 3/n)***.
- **Elaborating on Core Contributions.** As requested by reviewers (eWdD, xCSn, M2ve), we have further elaborated on the design and novelty of our framework's key components.
    - **Our Action:** We clarified that our Progress Reward stems from a **novel turn-level state evaluator** we built (as it's not provided by benchmarks) and that our curriculum is a principled, experimentally-validated design.
    - *This is supported by ablations in **Figure 4b & Table 3** and detailed in our responses to **R-eWdD (2/n), R-xCSn (2/n), and R-M2ve (1/n)***.

For other detailed questions, we have provided answers in our specific responses, which the AC can refer to.

**Positive early feedback from the reviewers:**

Our detailed responses and substantial manuscript updates were well-received, leading to an overall positive assessment. **Reviewer p3Yu raised their score from 4 to 6**, stating that our response **"addresses most of my concerns."** The other three reviewers (eWdD, xCSn, M2ve), who were already positive with initial scores of 6, maintained their ratings, with R-xCSn noting our response **"addressed most of my concerns"** and R-M2ve providing a final positive judgment.

We are confident that the rebuttal process has significantly strengthened our paper, and we hope you will find the revised version compelling.

Thank you for your time and consideration.

Sincerely,

The Authors of Submission 1928

---

### Meta-Review · Area_Chair_oVU8 · 2026-01-07

**Summary:**

This paper introduces **Environment Tuning**, a novel training paradigm for LLM agents designed to overcome the scarcity of expert trajectories and the instability of standard reinforcement learning (RL) in complex multi-turn tool-use tasks. Instead of relying on static supervised fine-tuning (SFT), which often leads to overfitting, the proposed framework enables agents to learn through dynamic environment exploration.
The method integrates three core components:

- **Actionable Environment Augmentation**: Automated feedback that provides corrective hints to guide the agent.

- **Structured Curriculum**: A four-stage learning roadmap that builds foundational skills before tackling complex exceptions.

- **Fine-grained Progress Rewards**: A turn-level evaluator that provides dense signals to stabilize exploration.

Using only 400 problem instances from the Berkeley Function-Calling Leaderboard (BFCL), the approach achieves competitive in-distribution performance and demonstrates superior out-of-distribution (OOD) generalization across diverse benchmarks, including ACEBench and -bench. The work is recognized by all reviewers for its innovative shift toward environment-based agent training. After the rebuttal, most of the concerns are addressed, so the area chair recommend a **acceptance** for this work.

**Reviewer Concerns:**

The area chair believes the following concerns were successfully addressed by the rebuttal:

- **Baseline and Generalization**: Authors added comparisons with **ToolRL (GRPO)** and expanded OOD testing to **-bench** , demonstrating consistent superiority over standard RL.

- **Scalability and Manual Effort**: Authors clarified that feedback is generated via an **automated LLM-based "Generator-Judge" workflow** , requiring only minimal "seed" examples rather than extensive manual engineering.

- **Real-world Applicability**: The mechanism was revealed to be a **non-invasive external wrapper** , making it portable to real-world APIs without modifying their underlying logic.

- **Reward Signal Novelty**: Authors clarified that the **Progress Reward** is derived from a newly built turn-level evaluator , providing dense signals not present in standard benchmarks.

The following concerns remain partially outstanding:

- **System Complexity**: The multi-component pipeline (wrapper, generator, judge) is perceived as potentially **cumbersome** with risks of error propagation.

- **Credit Assignment**: The average reward mechanism may not sufficiently punish bad actions at the very end of a trajectory. However, this is accepted as a trade-off for making complex tasks **tractable**.

**Reviewer Scores:**

Based on the rebuttal outcomes and the final comments provided by the reviewers, the assessment of potential score changes is as follows:

- Reviewer p3Yu: This reviewer already increased their score from 4 to 6 on 24 Nov before the incident.

- Reviewer eWdD: Originally at a 6, this reviewer maintained their score after the rebuttal.

- Reviewer xCSn: This reviewer maintained a score of 6.

- Reviewer M2ve: This reviewer maintained their score of 6. However, they noted in their final judgment that the paper is "between 6 and 8".

---

### Decision · Program_Chairs · 2026-01-26

Accept (Poster)